# Defending Against Physically Realizable Attacks on Image Classification

**Tong Wu, Liang Tong & Yevgeniy Vorobeychik**
Department of Computer Science and Engineering
Washington University in St. Louis
{tongwu, liangtong, yvorobeychik}@wustl.edu

## Abstract

We study the problem of defending deep neural network approaches for image classification from physically realizable attacks. First, we demonstrate that the two most scalable and effective methods for learning robust models, adversarial training with PGD attacks and randomized smoothing, exhibit limited effectiveness against three of the highest profile physical attacks. Next, we propose a new abstract adversarial model, rectangular occlusion attacks, in which an adversary places a small adversarially crafted rectangle in an image, and develop two approaches for efficiently computing the resulting adversarial examples. Finally, we demonstrate that adversarial training using our new attack yields image classification models that exhibit high robustness against the physically realizable attacks we study, offering the first effective generic defense against such attacks. [1]

## 1 Introduction

State-of-the-art effectiveness of deep neural networks has made it the technique of choice in a variety of fields, including computer vision (He et al., 2016), natural language processing (Sutskever et al., 2014), and speech recognition (Hinton et al., 2012). However, there have been a myriad of demonstrations showing that deep neural networks can be easily fooled by carefully perturbing pixels in an image through what have become known as *adversarial example attacks* (Szegedy et al., 2014; Goodfellow et al., 2015; Carlini & Wagner, 2017b; Vorobeychik & Kantarcioglu, 2018). In response, a large literature has emerged on *defending* deep neural networks against adversarial examples, typically either proposing techniques for learning more robust neural network models (Wong & Kolter, 2018; Wong et al., 2018; Raghunathan et al., 2018b; Cohen et al., 2019; Madry et al., 2018), or by detecting adversarial inputs (Metzen et al., 2017; Xu et al., 2018).

Particularly concerning, however, have been a number of demonstrations that implement adversarial perturbations directly in physical objects that are subsequently captured by a camera, and then fed through the deep neural network classifier (Boloor et al., 2019; Eykholt et al., 2018; Athalye et al., 2018b; Brown et al., 2018; Bhattad et al., 2020). Among the most significant of such *physical attacks* on deep neural networks are three that we specifically consider here: 1) the attack which fools face recognition by using adversarially designed eyeglass frames (Sharif et al., 2016), 2) the attack which fools stop sign classification by adding adversarially crafted stickers (Eykholt et al., 2018), and 3) the universal adversarial patch attack, which causes targeted misclassification of any object with the adversarially designed sticker (patch) (Brown et al., 2018). Oddly, while considerable attention has been devoted to defending against adversarial perturbation attacks in the digital space, there are no effective methods specifically to defend against such physical attacks.

Our first contribution is an empirical evaluation of the effectiveness of conventional approaches to robust ML against two physically realizable attacks: the eyeglass frame attack on face recognition (Sharif et al., 2016) and the sticker attack on stop signs (Eykholt et al., 2018). Specifically, we study the performance on adversarial training and randomized smoothing against these attacks, and show that both have limited effectiveness in this context (quite ineffective in some settings, and somewhat more effective, but still not highly robust, in others), despite showing moderate effectiveness against $l_\infty$ and $l_2$ attacks, respectively.

---

[1] The code can be found at https://github.com/tongwu2020/phattacks

Our second contribution is a novel abstract attack model which more directly captures the nature of common physically realizable attacks than the conventional $l_p$-based models. Specifically, we consider a simple class of *rectangular occlusion attacks* in which the attacker places a rectangular sticker onto an image, with both the location and the content of the sticker adversarially chosen. We develop several algorithms for computing such adversarial occlusions, and use adversarial training to obtain neural network models that are robust to these. We then experimentally demonstrate that our proposed approach is significantly more robust against physical attacks on deep neural networks than adversarial training and randomized smoothing methods that leverage $l_p$-based attack models.

**Related Work** While many approaches for defending deep learning in vision applications have been proposed, robust learning methods have been particularly promising, since alternatives are often defeated soon after being proposed (Madry et al., 2018; Raghunathan et al., 2018a; Wong & Kolter, 2018; Vorobeychik & Kantarcioglu, 2018). The standard solution approach for this problem is an adaptation of Stochastic Gradient Descent (SGD) where gradients are either with respect to the loss at the optimal adversarial perturbation for each $i$ (or approximation thereof, such as using heuristic local search (Goodfellow et al., 2015; Madry et al., 2018) or a convex over-approximation (Raghunathan et al., 2018b; Wang et al., 2018)), or with respect to the dual of the convex relaxation of the attacker maximization problem (Raghunathan et al., 2018a; Wong & Kolter, 2018; Wong et al., 2018). Despite these advances, adversarial training a la Madry et al. (2018) remains the most practically effective method for hardening neural networks against adversarial examples with $l_\infty$-norm perturbation constraints. Recently, randomized smoothing emerged as another class of techniques for obtaining robustness (Lecuyer et al., 2019; Cohen et al., 2019), with the strongest results in the context of $l_2$-norm attacks. In addition to training neural networks that are robust by construction, a number of methods study the problem of detecting adversarial examples (Metzen et al., 2017; Xu et al., 2018), with mixed results (Carlini & Wagner, 2017a). Of particular interest is recent work on detecting physical adversarial examples (Chou et al., 2018). However, detection is inherently weaker than robustness, which is our goal, as even perfect detection does not resolve the question of how to make decisions on adversarial examples. Finally, our work is in the spirit of other recent efforts that characterize robustness of neural networks to physically realistic perturbations, such as translations, rotations, blurring, and contrast (Engstrom et al., 2019; Hendrycks & Dietterich, 2019).

## 2 BACKGROUND

### 2.1 ADVERSARIAL EXAMPLES IN THE DIGITAL AND PHYSICAL WORLD

Adversarial examples involve modifications of input images that are either invisible to humans, or unsuspicious, and that cause systematic misclassification by state-of-the-art neural networks (Szegedy et al., 2014; Goodfellow et al., 2015; Vorobeychik & Kantarcioglu, 2018). Commonly, approaches for generating adversarial examples aim to solve an optimization problem of the following form:

$$\arg\max_{\boldsymbol{\delta}} L(f(\boldsymbol{x} + \boldsymbol{\delta}; \boldsymbol{\theta}), y) \qquad s.t. \|\boldsymbol{\delta}\|_p \leq \epsilon, \tag{1}$$

where $\boldsymbol{x}$ is the original input image, $\boldsymbol{\delta}$ is the adversarial perturbation, $L(\cdot)$ is the adversary's utility function (for example, the adversary may wish to maximize the cross-entropy loss), and $\| \cdot \|_p$ is some $l_p$ norm. While a host of such *digital attacks* have been proposed, two have come to be viewed as state of the art: the attack developed by Carlini & Wagner (2017b), and the projected gradient descent attack (PGD) by Madry et al. (2018).

While most of the work to date has been on attacks which modify the digital image directly, we focus on a class of *physical attacks* which entail modifying the actual object being photographed in order to fool the neural network that subsequently takes its digital representation as input. The attacks we will focus on will have three characteristics:

1. The attack can be implemented in the physical space (e.g., modifying the stop sign);
2. the attack has low *suspiciousness*; this is operationalized by modifying only a small part of the object, with the modification similar to common "noise" that obtains in the real world; for example, stickers on a stop sign would appear to most people as vandalism, but covering the stop sign with a printed poster would look highly suspicious; and
3. the attack causes misclassification by state-of-the-art deep neural network.

Since our ultimate purpose is defense, we will not concern ourselves with the issue of actually implementing the physical attacks. Instead, we will consider the digital representation of these attacks, ignoring other important issues, such as robustness to many viewpoints and printability. For example, in the case where the attack involves posting stickers on a stop sign, we will only be concerned with *simulating such stickers on digital images* of stop signs. For this reason, we refer to such attacks *physically realizable attacks*, to allude to the fact that it is possible to realize them in practice. It is evident that physically realizable attacks represent a somewhat stronger adversarial model than their actual implementation in the physical space. Henceforth, for simplicity, we will use the terms *physical attacks* and *physically realizable attacks* interchangeably.

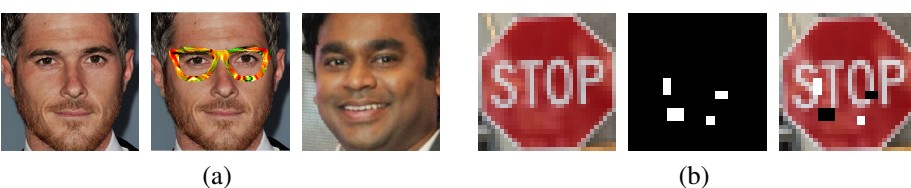

(a)  (b)

Figure 1: (a) An example of the eyeglass frame attack. Left: original face input image. Middle: modified input image (adversarial eyeglasses superimposed on the face). Right: an image of the predicted individual with the adversarial input in the middle image. (b) An example of the stop sign attack. Left: original stop sign input image. Middle: adversarial mask. Right: stop sign image with adversarial stickers, classified as a speed limit sign.

We consider three physically realizable attacks. The first is the attack on face recognition by Sharif et al. (2016), in which the attacker adds adversarial noise inside printed eyeglass frames that can subsequently be put on to fool the deep neural network (Figure 1a). The second attack posts adversarially crafted stickers on a stop sign to cause it to be misclassified as another road sign, such as the speed limit sign (Figure 1b) (Eykholt et al., 2018). The third, *adversarial patch*, attack designs a patch (a sticker) with adversarial noise that can be placed onto an arbitrary object, causing that object to be misclassified by a deep neural network (Brown et al., 2018).

## 2.2 Adversarially Robust Deep Learning

While numerous approaches have been proposed for making deep learning robust, many are heuristic and have soon after been defeated by more sophisticated attacks (Carlini & Wagner, 2017b; He et al., 2017; Carlini & Wagner, 2017a; Athalye et al., 2018a). Consequently, we focus on principled approaches for defense that have not been broken. These fall broadly into two categories: robust learning and randomized smoothing. We focus on a state-of-the-art representative from each class.

**Robust Learning** The goal of robust learning is to minimize a robust loss, defined as follows:

$$\boldsymbol{\theta}^* = \arg\min_{\boldsymbol{\theta}} \mathbb{E}_{(\boldsymbol{x},y)\sim\mathcal{D}} \left[ \max_{\|\boldsymbol{\delta}\|_p \leq \epsilon} L(f(\boldsymbol{x}+\boldsymbol{\delta};\boldsymbol{\theta}),y) \right], \tag{2}$$

where $\mathcal{D}$ denotes the training data set. In itself this is a highly intractable problem. Several techniques have been developed to obtain approximate solutions. Among the most effective in practice is the adversarial training approach by Madry et al. (2018), who use the PGD attack as an approximation to the inner optimization problem, and then take gradient descent steps with respect to the associated adversarial inputs. In addition, we consider a modified version of this approach termed *curriculum adversarial training* (Cai et al., 2018). Our implementation of this approach proceeds as follows: first, apply adversarial training for a small $\epsilon$, then increase $\epsilon$ and repeat adversarial training, and so on, increasing $\epsilon$ until we reach the desired level of adversarial noise we wish to be robust to.

**Randomized Smoothing** The second class of techniques we consider works by adding noise to inputs at both training and prediction time. The key idea is to construct a smoothed classifier $g(\cdot)$ from a base classifier $f(\cdot)$ by perturbing the input $\boldsymbol{x}$ with isotropic Gaussian noise with variance $\boldsymbol{\sigma}$. The prediction is then made by choosing a class with the highest probability measure with respect to the induced distribution of $f(\cdot)$ decisions:

$$g(\boldsymbol{x}) = \arg\max_c P(f(\boldsymbol{x}+\boldsymbol{\sigma})=c), \quad \boldsymbol{\sigma} \sim \mathcal{N}\left(0,\sigma^2 I\right). \tag{3}$$

To achieve provably robust classification in this manner one typically trains the classifier $f(\cdot)$ by adding Gaussian noise to inputs at training time (Lecuyer et al., 2019; Cohen et al., 2019).

# 3 ROBUSTNESS OF CONVENTIONAL ROBUST ML METHODS AGAINST PHYSICAL ATTACKS

Most of the approaches for endowing deep learning with adversarial robustness focus on adversarial models in which the attacker introduces $l_p$-bounded adversarial perturbations over the entire input. Earlier we described two representative approaches in this vein: *adversarial training*, commonly focused on robustness against $l_\infty$ attacks, and *randomized smoothing*, which is most effective against $l_2$ attacks (although certification bounds can be extended to other $l_p$ norms as well). We call these methods *conventional robust ML*.

In this section, we ask the following question:

*Are conventional robust ML methods robust against physically realizable attacks?*

This is similar to the question was asked in the context of malware classifier evasion by Tong et al. (2019), who found that $l_p$-based robust ML methods can indeed be successful in achieving robustness against realizable evasion attacks. Ours is the first investigation of this issue in computer vision applications and for deep neural networks, where attacks involve adversarial masking of objects.[2]

We study this issue experimentally by considering two state-of-the-art approaches for robust ML: adversarial training a-la-Madry et al. (2018), along with its curriculum learning variation (Cai et al., 2018), and randomized smoothing, using the implementation by Cohen et al. (2019). These approaches are applied to defend against two physically realizable attacks described in Section 2.1: an attack on face recognition which adds adversarial eyeglass frames to faces (Sharif et al., 2016), and an attack on stop sign classification which adds adversarial stickers to a stop sign to cause misclassification (Eykholt et al., 2018).

We consider several variations of adversarial training, as a function of the $l_\infty$ bound, $\epsilon$, imposed on the adversary. Just as Madry et al. (2018), adversarial instances in adversarial training were generated using PGD. We consider attacks with $\epsilon \in \{4, 8\}$ (adversarial training failed to make progress when we used $\epsilon = 16$). For curriculum adversarial training, we first performed adversarial training with $\epsilon = 4$, then doubled $\epsilon$ to 8 and repeated adversarial training with the model robust to $\epsilon = 4$, then doubled $\epsilon$ again, and so on. In the end, we learned models for $\epsilon \in \{4, 8, 16, 32\}$. For all versions of adversarial training, we consider 7 and 50 iterations of the PGD attack. We used the learning rate of $\epsilon/4$ for the former and 1 for the latter. In all cases, pixels are in $0 \sim 255$ range and retraining was performed for 30 epochs using the ADAM optimizer.

For randomized smoothing, we consider noise levels $\sigma \in \{0.25, 0.5, 1\}$ as in Cohen et al. (2019), and take 1,000 Monte Carlo samples at test time.

## 3.1 ADVERSARIAL EYEGLASSES IN FACE RECOGNITION

We applied white-box dodging (untargeted) attacks on the face recognition systems (FRS) from Sharif et al. (2016). We used both the VGGFace data and transferred VGGFace CNN model for the face recognition task, subselecting 10 individuals, with 300-500 face images for each. Further details about the dataset, CNN architecture, and training procedure are in Appendix A. For the attack, we used identical frames as in Sharif et al. (2016) occupying 6.5% of the pixels. Just as Sharif et al. (2016), we compute attacks (that is, adversarial perturbations inside the eyeglass frame area) by using the learning rate 20 as well as momentum value 0.4, and vary the number of attack iterations between 0 (no attack) and 300.

Figure 2 presents the results of classifiers obtained from adversarial training (left) as well as curriculum adversarial training (middle), in terms of accuracy (after the attack) as a function of the number of iterations of the Sharif et al. (2016) eyeglass frame attack. First, it is clear that none of the variations of adversarial training are particularly effective once the number of physical attack iterations is

---

[2]Several related efforts study robustness of deep neural networks to other variants of physically realistic perturbations (Engstrom et al., 2019; Hendrycks & Dietterich, 2019).

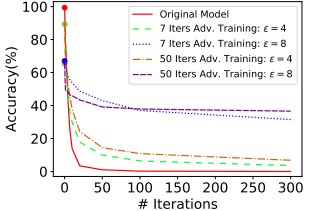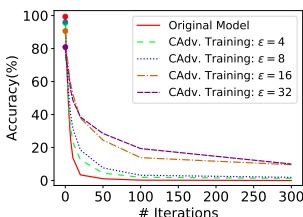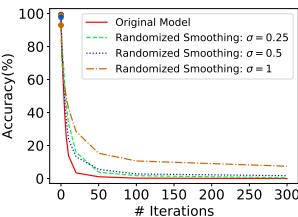

Figure 2: Performance of adversarial training (left), curriculum adversarial training (with 7 PGD iterations) (middle), and randomized smoothing (right) against the eyeglass frame attack.

above 20. The best performance in terms of adversarial robustness is achieved by adversarial training with $\epsilon = 8$, for approaches using either 7 or 50 PGD iterations (the difference between these appears negligible). However, non-adversarial accuracy for these models is below 70%, a $\sim$20% drop in accuracy compared to the original model. Moreover, adversarial accuracy is under 40% for sufficiently strong physical attacks. Curriculum adversarial training generally achieves significantly higher non-adversarial accuracy, but is far less robust, even when trained with PGD attacks that use $\epsilon = 32$.

Figure 2 (right) shows the performance of randomized smoothing when faced with the eyeglass frames attack. It is readily apparent that randomized smoothing is ineffective at deflecting this physical attack: even as we vary the amount of noise we add, accuracy after attacks is below 20% even for relatively weak attacks, and often drops to nearly 0 for sufficiently strong attacks.

## 3.2 Adversarial Stickers on Stop Signs

Following Eykholt et al. (2018), we use the LISA traffic sign dataset for our experiments, and 40 stop signs from this dataset as our test data and perform untargeted attacks (this is in contrast to the original work, which is focused on targeted attacks). For the detailed description of the data and the CNN used for traffic sign prediction, see Appendix A. We apply the same settings as in the original attacks and use ADAM optimizer with the same parameters. Since we observed few differences in performance between running PGD for 7 vs. 50 iterations, adversarial training methods in this section all use 7 iterations of PGD.

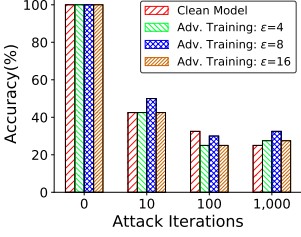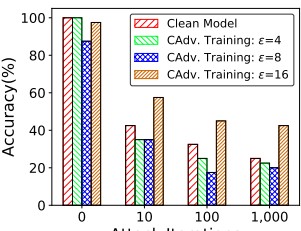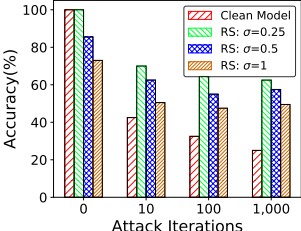

Figure 3: Performance of adversarial training (left), curriculum adversarial training (with 7 PGD iterations) (middle), and randomized smoothing (right) against the stop sign attack.

Again, we begin by considering adversarial training (Figure 3, left and middle). In this case, both the original and curriculum versions of adversarial training with PGD are ineffective when $\epsilon = 32$ (error rates on clean data are above 90%); these are consequently omitted from the plots. Curriculum adversarial training with $\epsilon = 16$ has the best performance on adversarial data, and works well on clean data. Surprisingly, most variants of adversarial training perform at best marginally better than the original model against the stop sign attack. Even the best variant has relatively poor performance, with robust accuracy under 50% for stronger attacks.

Figure 3 (right) presents the results for randomized smoothing. In this set of experiments, we found that randomized smoothing performs inconsistently. To address this, we used 5 random seeds to repeat the experiments, and use the resulting mean values in the final results. Here, the best variant

uses $\sigma = 0.25$, and, unlike experiments with the eyeglass frame attack, significantly outperforms adversarial training, reaching accuracy slightly above 60% even for the stronger attacks. Nevertheless, even randomized smoothing results in significant degradation of effectiveness on adversarial instances (nearly 40%, compared to clean data).

## 3.3 DISCUSSION

There are two possible reasons why conventional robust ML perform poorly against physical attacks: 1) adversarial models involving $l_p$-bounded perturbations are too hard to enable effective robust learning, and 2) the conventional attack model is too much of a mismatch for realistic physical attacks. In Appendix B, we present evidence supporting the latter. Specifically, we find that conventional robust ML models exhibit much higher robustness when faced with the $l_p$-bounded attacks they are trained to be robust to.

## 4 PROPOSED APPROACH: DEFENSE AGAINST OCCLUSION ATTACKS (DOA)

As we observed in Section 3, conventional models for making deep learning robust to attack can perform quite poorly when confronted with physically realizable attacks. In other words, the evidence strongly suggests that the conventional models of attacks in which attackers can make $l_p$-bounded perturbations to input images are not particularly useful if one is concerned with the main physical threats that are likely to be faced in practice. However, given the diversity of possible physical attacks one may perpetrate, is it even possible to have a meaningful approach for ensuring robustness against a broad range of physical attacks? For example, the two attacks we considered so far couldn't be more dissimilar: in one, we engineer eyeglass frames; in another, stickers on a stop sign. We observe that the key common element in these attacks, and many other physical attacks we may expect to encounter, is that they involve the introduction of *adversarial occlusions* to a part of the input. The common constraint faced in such attacks is to avoid being suspicious, which effectively limits the size of the adversarial occlusion, but not necessarily its shape or location. Next, we introduce a simple abstract model of occlusion attacks, and then discuss how such attacks can be computed and how we can make classifiers robust to them.

## 4.1 ABSTRACT ATTACK MODEL: RECTANGULAR OCCLUSION ATTACKS (ROA)

We propose the following simple abstract model of adversarial occlusions of input images. The attacker introduces a fixed-dimension rectangle. This rectangle can be placed by the adversary anywhere in the image, and the attacker can furthermore introduce $l_\infty$ noise *inside the rectangle* with an exogenously specified high bound $\epsilon$ (for example, $\epsilon = 255$, which effectively allows addition of arbitrary adversarial noise). This model bears some similarity to $l_0$ attacks, but the rectangle imposes a contiguity constraint, which reflects common physical limitations. The model is clearly abstract: in practice, for example, adversarial occlusions need not be rectangular or have fixed dimensions (for example, the eyeglass frame attack is clearly not rectangular), but at the same time cannot usually be arbitrarily superimposed on an image, as they are implemented in the physical environment. Nevertheless, the model reflects some of the most important aspects common to many physical attacks, such as stickers placed on an adversarially chosen portion of the object we wish to identify. We call our attack model a *rectangular occlusion attack (ROA)*. An important feature of this attack is that it is untargeted: since our ultimate goal is to defend against physical attacks whatever their target, considering untargeted attacks obviates the need to have precise knowledge about the attacker's goals. For illustrations of the ROA attack, see Appendix C.

## 4.2 COMPUTING ATTACKS

The computation of ROA attacks involves 1) identifying a region to place the rectangle in the image, and 2) generating fine-grained adversarial perturbations restricted to this region. The former task can be done by an exhaustive search: consider all possible locations for the upper left-hand corner of the rectangle, compute adversarial noise inside the rectangle using PGD for each of these, and choose the worst-case attack (i.e., the attack which maximizes loss computed on the resulting image). However, this approach would be quite slow, since we need to perform PGD inside the rectangle for every possible position. Our approach, consequently, decouples these two tasks. Specifically, we first

perform an exhaustive search using a grey rectangle to find a position for it that maximizes loss, and then fix the position and apply PGD inside the rectangle.

An important limitation of the exhaustive search approach for ROA location is that it necessitates computations of the loss function for every possible location, which itself requires full forward propagation each time. Thus, the search itself is still relatively slow. To speed the process up further, we use the gradient of the input image to identify candidate locations. Specifically, we select a subset of $C$ locations for the sticker with the highest magnitude of the gradient, and only exhaustively search among these $C$ locations. $C$ is exogenously specified to be small relative to the number of pixels in the image, which significantly limits the number of loss function evaluations. Full details of our algorithms for computing ROA are provided in Appendix D.

### 4.3 DEFENDING AGAINST ROA

Once we are able to compute the ROA attack, we apply the standard adversarial training approach for defense. We term the resulting classifiers robust to our abstract adversarial occlusion attacks *Defense against Occlusion Attacks (DOA)*, and propose these as an alternative to conventional robust ML for defending against physical attacks. As we will see presently, this defense against ROA is quite adequate for our purposes.

## 5 EFFECTIVENESS OF DOA AGAINST PHYSICALLY REALIZABLE ATTACKS

We now evaluate the effectiveness of DOA—that is, adversarial training using the ROA threat model we introduced—against physically realizable attacks (see Appendix G for some examples that defeat conventional methods but not DOA). Recall that we consider only digital representations of the corresponding physical attacks. Consequently, we can view our results in this section as a lower bound on robustness to actual physical attacks, which have to deal with additional practical constraints, such as being robust to multiple viewpoints. In addition to the two physical attacks we previously considered, we also evaluate DOA against the *adversarial patch* attack, implemented on both face recognition and traffic sign data.

### 5.1 DOA AGAINST ADVERSARIAL EYEGLASSES

We consider two rectangle dimensions resulting in comparable area: $100 \times 50$ and $70 \times 70$, both in pixels. Thus, the rectangles occupy approximately 10% of the $224 \times 224$ face images. We used $\{30, 50\}$ iterations of PGD with $\epsilon = 255/2$ to generate adversarial noise inside the rectangle, and with learning rate $\alpha = \{8, 4\}$ correspondingly. For the gradient version of ROA, we choose $C = 30$. DOA adversarial training is performed for 5 epochs with a learning rate of 0.0001.

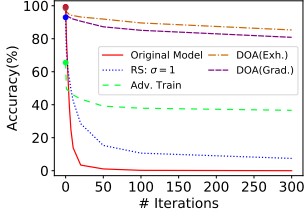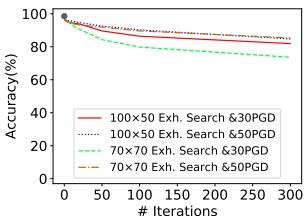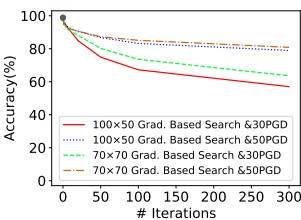

Figure 4: Performance of DOA (using the $100 \times 50$ rectangle) against the eyeglass frame attack in comparison with conventional methods. Left: comparison between DOA, adversarial training, and randomized smoothing (using the most robust variants of these). Middle/Right: Comparing DOA performance for different rectangle dimensions and numbers of PGD iterations inside the rectangle. Middle: using exhaustive search for ROA; right: using the gradient-based heuristic for ROA.

Figure 4 (left) presents the results comparing the effectiveness of DOA against the eyeglass frame attack on face recognition to adversarial training and randomized smoothing (we took the most robust variants of both of these). We can see that DOA yields significantly more robust classifiers for this domain. The gradient-based heuristic does come at some cost, with performance slightly worse

than when we use exhaustive search, but this performance drop is relatively small, and the result is still far better than conventional robust ML approaches. Figure 4 (middle and right) compares the performance of DOA between two rectangle variants with different dimensions. The key observation is that as long as we use enough iterations of PGD inside the rectangle, changing its dimensions (keeping the area roughly constant) appears to have minimal impact.

## 5.2  DOA AGAINST THE STOP SIGN ATTACK

We now repeat the evaluation with the traffic sign data and the stop sign attack. In this case, we used $10 \times 5$ and $7 \times 7$ rectangles covering $\sim$5 % of the $32 \times 32$ images. We set $C = 10$ for the gradient-based ROA. Implementation of DOA is otherwise identical as in the face recognition experiments above.

We present our results using the square rectangle, which in this case was significantly more effective; the results for the $10 \times 5$ rectangle DOA attacks are in Appendix F. Figure 5 (left) compares the effectiveness of DOA against the stop sign attack on traffic sign data with the best variants of adversarial training and randomized smoothing. Our results here are for 30 iterations of PGD; in Appendix F, we study the impact of varying the number of PGD iterations. We can observe that DOA is again significantly more robust, with robust accuracy over 90% for the exhaustive search variant, and $\sim$85% for the gradient-based variant, even for stronger attacks. Moreover, DOA remains 100% effective at classifying stop signs on clean data, and exhibits $\sim$95% accuracy on the full traffic sign classification task.

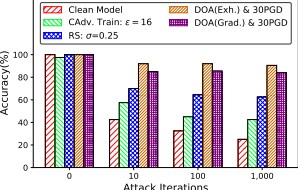 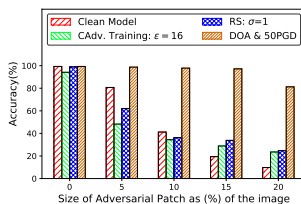

Figure 5: Performance of the best case of adversarial training, randomized smoothing, and DOA against the stop sign attack (left) and adversarial patch attack (right).

## 5.3  DOA AGAINST ADVERSARIAL PATCH ATTACKS

Finally, we evaluate DOA against the adversarial patch attacks. In these attacks, an adversarial patch (e.g., sticker) is designed to be placed on an object with the goal of inducing a target prediction. We study this in both face recognition and traffic sign classification tasks. Here, we present the results for face recognition; further detailed results on both datasets are provided in Appendix F.

As we can see from Figure 5 (right), adversarial patch attacks are quite effective once the attack region (fraction of the image) is 10% or higher, with adversarial training and randomized smoothing both performing rather poorly. In contrast, DOA remains highly robust even when the adversarial patch covers 20% of the image.

## 6  CONCLUSION

As we have shown, conventional methods for making deep learning approaches for image classification robust to physically realizable attacks tend to be relatively ineffective. In contrast, a new threat model we proposed, rectangular occlusion attacks (ROA), coupled with adversarial training, achieves high robustness against several prominent examples of physical attacks. While we explored a number of variations of ROA attacks as a means to achieve robustness against physical attacks, numerous questions remain. For example, can we develop effective methods to certify robustness against ROA, and are the resulting approaches as effective in practice as our method based on a combination of heuristically computed attacks and adversarial training? Are there other types of occlusions that are more effective? Answers to these and related questions may prove a promising

path towards practical robustness of deep learning when deployed for downstream applications of computer vision such as autonomous driving and face recognition.

ACKNOWLEDGMENTS

This work was partially supported by the NSF (IIS-1905558, IIS-1903207), ARO (W911NF-19-1-0241), and NVIDIA.

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

# A   DESCRIPTION OF DATASETS AND DEEP LEARNING CLASSIFIERS

## A.1   FACE RECOGNITION

The VGGFace dataset[3] (Parkhi et al., 2015) is a benchmark for face recognition, containing 2622 subjusts with 2.6 million images in total. We chose ten subjects: A. J. Buckley, A. R. Rahman, Aamir Khan, Aaron Staton, Aaron Tveit, Aaron Yoo, Abbie Cornish, Abel Ferrara, Abigail Breslin, and Abigail Spencer, and subselected face images pertaining only to these individuals. Since approximately half of the images cannot be downloaded, our final dataset contains 300-500 images for each subject.

We used the standard corp-and-resize method to process the data to be $224 \times 224$ pixels, and split the dataset into training, validation, and test according to a 7:2:1 ratio for each subject. In total, the data set has 3178 images in the training set, 922 images in the validation set, and 470 images in the test set.

We use the VGGFace convolutional neural network (Parkhi et al., 2015) model, a variant of the VGG16 model containing 5 convolutional layer blocks and 3 fully connected layers. We make use of standard transfer learning as we only classify 10 subjects, keeping the convolutional layers as same as VGGFace structure,[4] but changing the fully connected layer to be $1024 \rightarrow 1024 \rightarrow 10$ instead of $4096 \rightarrow 4096 \rightarrow 2622$. Specifically, in our Pytorch implementation, we convert the images from RGB to BGR channel orders and subtract the mean value [129.1863, 104.7624, 93.5940] in order to use the pretrained weights from VGG-Face on convolutional layers. We set the batch size to be 64 and use Pytorch built-in Adam Optimizer with an initial learning rate of $10^{-4}$ and default parameters in Pytorch.[5] We drop the learning rate by 0.1 every 10 epochs. Additionally, we used validation set accuracy to keep track of model performance and choose a model in case of overfitting. After 30 epochs of training, the model successfully obtains 98.94 % on test data.

## A.2   TRAFFIC SIGN CLASSIFICATION

To be consistent with (Eykholt et al., 2018), we select the subset of LISA which contains 47 different U.S. traffic signs (Møgelmose et al., 2012). To alleviate the problem of imbalance and extremely blurry data, we picked 16 best quality signs with 3509 training and 1148 validation data points. From the validation data, we obtain the test data that includes only 40 stop signs to evaluate performance with respect to the stop sign attack, as done by Eykholt et al. (2018). In the main body of the paper, we present results only on this test data to evaluate robustness to stop sign attacks. In the appendix below, we also include performance on the full validation set without adversarial manipulation.

All the data was processed by standard crop-and-resize to $32 \times 32$ pixels. We use the LISA-CNN architecture defined in (Eykholt et al., 2018), and construct a convolutional neural network containing three convolutional layers and one fully connected layer. We use the Adam Optimizer with initial learning rate of $10^{-1}$ and default parameters [5], dropping the learning rate by 0.1 every 10 epochs. We set the batch size to be 128. After 30 epochs, we achieve the 98.69 % accuracy on the validation set, and 100% accuracy in identifying the stop signs in our test data.

---

[3] http://www.robots.ox.ac.uk/~vgg/data/vgg_face/.

[4] External code that we use for transfering VGG-Face to Pytorch Framework is available at https://github.com/prlz77/vgg-face.pytorch

[5] Default Pytorch Adam parameters stand for $\beta_1$=0.9, $\beta_1$=0.999 and $\epsilon$=$10^{-8}$

## B EFFECTIVENESS OF CONVENTIONAL ROBUST ML METHODS AGAINST $l_\infty$ AND $l_2$ ATTACKS

In this appendix, we show that adversarial training and randomized smoothing degrade more gracefully when faced with attacks that they are designed for. In particular, we consider here variants of projected gradient descent (PGD) for both the $l_\infty$ and $l_2$ attacks Madry et al. (2018). In particular, the form of PGD for the $l_\infty$ attack is

$$x_{t+1} = \text{Proj}(x_t + \alpha \text{sgn}(\nabla L(x_t; \theta))),$$

where $\text{Proj}$ is a projection operator which clips the result to be feasible, $x_t$ the adversarial example in iteration $t$, $\alpha$ the learning rate, and $L(\cdot)$ the loss function. In the case of an $l_2$ attack, PGD becomes

$$x_{t+1} = \text{Proj}\left(x_t + \alpha \frac{\nabla L(x_t; \theta)}{\|\nabla L(x_t; \theta)\|_2}\right),$$

where the projection operator normalizes the perturbation $\delta = x_{t+1} - x_t$ to have $\|\delta\|_2 \le \epsilon$ if it doesn't already Kolter & Madry (2019).

The experiments were done on the face recognition and traffic sign datasets, but unlike physical attacks on stop signs, we now consider adversarial perturbations to *all* sign images.

### B.1 FACE RECOGNITION

Table 1: Curriculum Adversarial Training against 7 Iterations $L_\infty$ Attacks on Face Recognition

|  | Attack Strength | | | | |
| --- | --- | --- | --- | --- | --- |
|  | $\epsilon = 0$ | $\epsilon = 2$ | $\epsilon = 4$ | $\epsilon = 8$ | $\epsilon = 16$ |
| Clean Model | **98.94%** | 57.87% | 13.62% | 0% | 0% |
| CAdv. Training: $\epsilon = 4$ | 97.45% | **94.68%** | 87.02% | 65.11% | 17.23% |
| CAdv. Training: $\epsilon = 8$ | 96.17% | 93.40% | **89.36%** | 75.53% | 31.49% |
| CAdv. Training: $\epsilon = 16$ | 90.64% | 88.09% | 84.04% | **75.96%** | 45.74% |
| CAdv. Training: $\epsilon = 32$ | 80.85% | 76.60% | 74.04% | 65.10% | **47.87%** |

Table 2: Curriculum Adversarial Training against 20 Iterations $L_\infty$ Attacks on Face Recognition

|  | Attack Strength | | | | |
| --- | --- | --- | --- | --- | --- |
|  | $\epsilon = 0$ | $\epsilon = 2$ | $\epsilon = 4$ | $\epsilon = 8$ | $\epsilon = 16$ |
| Clean Model | **98.94%** | 44.04% | 1.70% | 0% | 0% |
| CAdv. Training: $\epsilon = 4$ | 97.45% | **94.68%** | 85.74% | 46.60% | 5.11% |
| CAdv. Training: $\epsilon = 8$ | 96.17% | 93.19% | **88.94%** | 69.36% | 9.57% |
| CAdv. Training: $\epsilon = 16$ | 90.64% | 88.08% | 83.83% | **73.62%** | 35.96% |
| CAdv. Training: $\epsilon = 32$ | 80.85% | 76.17% | 74.04% | 63.82% | **43.62%** |

We begin with our results on the face recognition dataset. Tables 1 and 2 present results for (curriculum) adversarial training for varying $\epsilon$ of the $l_\infty$ attacks, separately for training and evaluation. As we can see, curriculum adversarial training with $\epsilon = 16$ is generally the most robust, and remains reasonably effective for relatively large perturbations. However, we do observe a clear tradeoff between accuracy on non-adversarial data and robustness, as one would expect.

Table 3: Randomized Smoothing against 20 Iterations $L_2$ Attacks on Face Recognition

| | Attack Strength | | | | | | |
|---|---|---|---|---|---|---|---|
| | $\epsilon = 0$ | $\epsilon = 0.5$ | $\epsilon = 1$ | $\epsilon = 1.5$ | $\epsilon = 2$ | $\epsilon = 2.5$ | $\epsilon = 3$ |
| Clean Model | **98.94%** | 93.19% | 70.85% | 44.68% | 22.13% | 8.29% | 3.19% |
| RS: $\sigma = 0.25$ | 98.51% | **97.23%** | **95.53%** | **91.70%** | 81.06% | 67.87% | 52.97% |
| RS: $\sigma = 0.5$ | 97.65% | 94.25% | 93.61% | **91.70%** | 87.87% | 82.55% | 71.70% |
| RS: $\sigma = 1$ | 92.97% | 93.19% | 91.70% | 91.06% | **88.51%** | **85.53%** | **82.98%** |

Table 3 presents the results of using randomized smoothing on face recognition data, when facing the $l_2$ attacks. Again, we observe a high level of robustness and, in most cases, relatively limited drop in performance, with $\sigma = 0.5$ perhaps striking the best balance.

## B.2 TRAFFIC SIGN CLASSIFICATION

Table 4: Curriculum Adversarial Training against 7 Iterations $L_\infty$ Attacks on Traffic Signs

| | Attack Strength | | | | |
|---|---|---|---|---|---|
| | $\epsilon = 0$ | $\epsilon = 2$ | $\epsilon = 4$ | $\epsilon = 8$ | $\epsilon = 16$ |
| Clean Model | 98.69% | 90.24% | 65.24% | 33.80% | 10.10% |
| CAdv. Training: $\epsilon = 4$ | **99.13%** | **97.13%** | 93.47% | 63.85% | 20.73% |
| CAdv. Training: $\epsilon = 8$ | 98.72% | 96.86% | **93.90%** | 81.70% | 38.24% |
| CAdv. Training: $\epsilon = 16$ | 96.95% | 95.03% | 92.77% | **87.63%** | **64.02%** |
| CAdv. Training: $\epsilon = 32$ | 65.63% | 53.83% | 50.87% | 46.69% | 38.07% |

Table 5: Curriculum Adversarial Training against 20 Iterations $L_\infty$ Attacks on Traffic Signs

| | Attack Strength | | | | |
|---|---|---|---|---|---|
| | $\epsilon = 0$ | $\epsilon = 2$ | $\epsilon = 4$ | $\epsilon = 8$ | $\epsilon = 16$ |
| Clean Model | 98.69% | 89.54% | 61.58% | 24.65% | 5.14% |
| CAdv. Training: $\epsilon = 4$ | **99.13%** | **96.95%** | 91.90% | 56.53% | 12.02% |
| CAdv. Training: $\epsilon = 8$ | 98.72% | 96.68% | **93.64%** | 76.22% | 28.13% |
| CAdv. Training: $\epsilon = 16$ | 96.95% | 95.03% | 92.51% | **86.76%** | **54.01%** |
| CAdv. Training: $\epsilon = 32$ | 65.63% | 53.83% | 50.78% | 46.08% | 36.49% |

Tables 4 and 5 present evaluation on traffic sign data for curriculum adversarial training against the $l_\infty$ attack for varying $\epsilon$. As with face recognition data, we can observe that the approaches tend to be relatively robust, and effective on non-adversarial data for adversarial training methods using $\epsilon < 32$.

Table 6: Randomized Smoothing against 20 Iterations $L_2$ Attacks on Traffic Signs

| | Attack Strength | | | | | | |
|---|---|---|---|---|---|---|---|
| | $\epsilon = 0$ | $\epsilon = 0.5$ | $\epsilon = 1$ | $\epsilon = 1.5$ | $\epsilon = 2$ | $\epsilon = 2.5$ | $\epsilon = 3$ |
| Clean Model | **98.69%** | 61.67% | 25.78% | 11.50% | 7.84% | 4.97% | 3.57% |
| RS: $\sigma = 0.25$ | 98.22% | 89.08% | 55.69% | 34.06% | 23.46% | 18.61% | 14.75% |
| RS: $\sigma = 0.5$ | 96.28% | **90.80%** | **76.13%** | 52.64% | 35.31% | 23.43% | 16.52% |
| RS: $\sigma = 1$ | 88.21% | 83.68% | 75.49% | **64.90%** | **50.03%** | **36.53%** | **26.22%** |

The results of randomized smoothing on traffic sign data are given in Table 6. Since images are smaller here than in VGGFace, lower values of $\epsilon$ for the $l_2$ attacks are meaningful, and for $\epsilon \leq 1$ we

generally see robust performance on randomized smoothing, with $\sigma = 0.5$ providing a good balance between non-adversarial accuracy and robustness, just as before.

## C  EXAMPLES OF ROA

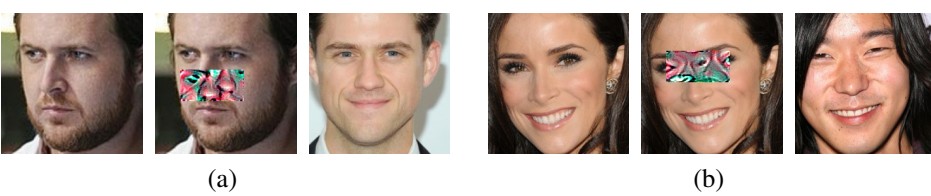

$\qquad\qquad\qquad$ (a) $\qquad\qquad\qquad\qquad\qquad\qquad$ (b)

Figure 6: Examples of the ROA attack on face recognition, using a rectangle of size $100 \times 50$. (a) Left: the original A. J. Buckley's image. Middle: modified input image (ROA superimposed on the face). Right: an image of the predicted individual who is Aaron Tveit with the adversarial input in the middle image. (b) Left: the original Abigail Spencer's image. Middle: modified input image (ROA superimposed on the face). Right: an image of the predicted individual who is Aaron Yoo with the adversarial input in the middle image.

Figure 6 provides several examples of the ROA attack in the context of face recognition. Note that in these examples, the adversaries choose to occlude the noise on upper lip and eye areas of the image, and, indeed, this makes the face more challenging to recognize even to a human observer.

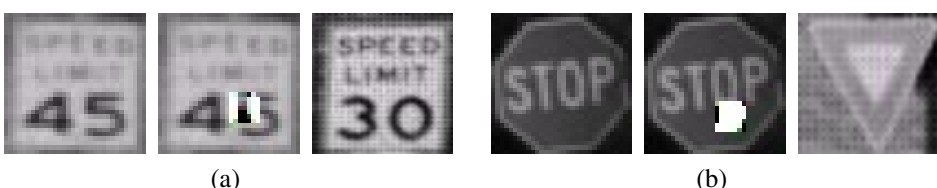

$\qquad\qquad\qquad$ (a) $\qquad\qquad\qquad\qquad\qquad\qquad$ (b)

Figure 7: Examples of the ROA attack on traffic sign, using a rectangle of size $7 \times 7$. (a) Left: the original Speedlimit45 sign. Middle: modified input image (ROA superimposed on the sign). Right: an image of the predicted which is Speedlimit30 with the adversarial input in the middle image. (b) Left: the original Stop sign. Middle: modified input image (ROA superimposed on the sign). Right: an image of the predicted Yield sign with the adversarial input in the middle image.

## D  DETAILED DESCRIPTION OF THE ALGORITHMS FOR COMPUTING THE RECTANGULAR OCCLUSION ATTACKS

Our basic algorithm for computing rectangular occlusion attacks (ROA) proceeds through the following two steps:

1. Iterate through possible positions for the rectangle's upper left-hand corner point in the image. Find the position for a grey rectangle (RGB value =[127.5,127.5,127.5]) in the image that maximizes loss.
2. Generate high-$\epsilon$ $l_\infty$ noise inside the rectangle at the position computed in step 1.

Algorithm 1 presents the full algorithm for identifying the ROA position, which amounts to exhaustive search through the image pixel region. This algorithm has several parameters. First, we assume that images are squares with dimensions $N^2$. Second, we introduce a *stride* parameter $S$. The purpose of this parameter is to make location computation faster by only considering every other $S$th pixel during the search (in other words, we skip $S$ pixels each time). For our implementation of ROA attacks, we choose the stride parameter $S = 5$ for face recognition and $S = 2$ for traffic sign classification.

---

**Algorithm 1** Computation of ROA position using exhaustive search.

    **Input:**
Data: $X_i, y_i$;    Test data shape: $N \times N$;    Target model parameters: $\theta$;    Stride: $S$ ;
    **Output:**
ROA Position: $(j', k')$
1. **function** ExhaustiveSearching($Model, X_i, y_i, N, S$)
2.   **for** j **in** range($N/S$) **do**:
3.     **for** k **in** range($N/S$) **do**:
4.       Generate the adversarial $X_i^{adv}$ image by:
5.       place a grey rectangle onto the image with top-left corner at $(j \times S, k \times S)$;
6.       **if** L($X_i^{adv}, y_i, \theta$) is higher than previous loss:
7.         **Update** $(j', k') = (j, k)$
8.     **end for**
9.   **end for**
10.  **return** $(j', k')$

---

**Algorithm 2** Computation of ROA position using gradient-based search.

    **Input:**
Data: $X_i, y_i$;    Test data shape: $N \times N$;    Target Model: $\theta$;    Stride: $S$ ;
Number of Potential Candidates: $C$;
    **Output:**
Best Sticker Position: $(j', k')$

1. **function** GradientBasedSearch($X_i, y_i, N, S, C, \theta$)
2.   Calculate the gradient $\nabla L$ of Loss($X_i, y_i, \theta$) w.r.t. $X_i$
3.   $\mathbb{J}, \mathbb{K}$ = HelperSearching($\nabla L, N, S, C$)
4.   **for** $j, k$ **in** $\mathbb{J}, \mathbb{K}$ **do**:
5.     Generate the adversarial $X_i^{adv}$ image by:
6.     put the sticker on the image where top-left corner at $(j \times S, k \times S)$;
7.     **if** Loss($X_i^{adv}, y_i, \theta$) is higher than previous loss:
8.       **Update** $(j', k') = (j, k)$
9.   **end for**
10.  **return** $(j', k')$

1. **function** HelperSearching($\nabla L, N, S, C$)
2.   **for** j **in** range($N/S$) **do**:
3.     **for** k **in** range($N/S$) **do**:
4.       Calculate the Sensitivity value $L = \sum_{i \in \text{rectangle}} (\nabla L_i)^2$ where top-left corner at $(j \times S, k \times S)$;
6.       **if** the Sensitivity value $L$ is in top $C$ of previous values:
7.         Put $(j, k)$ in $\mathbb{J}, \mathbb{K}$ and discard $(j_s, k_s)$ with lowest $L$
8.     **end for**
9.   **end for**
10.  **return** $\mathbb{J}, \mathbb{K}$

---

Despite introducing the tunable stride parameter, the search for the best location for ROA still entails a large number of loss function evaluations, which are somewhat costly (since each such evaluation means a full forward pass through the deep neural network), and these costs add up quickly. To speed things up, we consider using the magnitude of the gradient of the loss as a measure of sensitivity of particular regions to manipulation. Specifically, suppose that we compute a gradient $\nabla L$, and let $\nabla L_i$ be the gradient value for a particular pixel $i$ in the image. Now, we can iterate over the possible ROA locations, but for each location compute the gradient of the loss at that location corresponding to the rectangular region. We do this by adding squared gradient values $(\nabla L_i)^2$ over pixels $i$ in the rectangle. We use this approach to find the top $C$ candidate locations for the rectangle. Finally, we consider each of these, computing the actual loss for each location, to find the position of ROA. The full algorithm is provided as Algorithm 2.

Once we've found the place for the rectangle, our next step is to introduce adversarial noise inside it. For this, we use the $l_\infty$ version of the PGD attack, restricting perturbations to the rectangle. We used $\{7, 20, 30, 50\}$ iterations of PGD to generate adversarial noise inside the rectangle, and with learning rate $\alpha = \{32, 16, 8, 4\}$ correspondingly.

| Original image | Grad. plot | Grad. searching | Exh. searching |
|---|---|---|---|

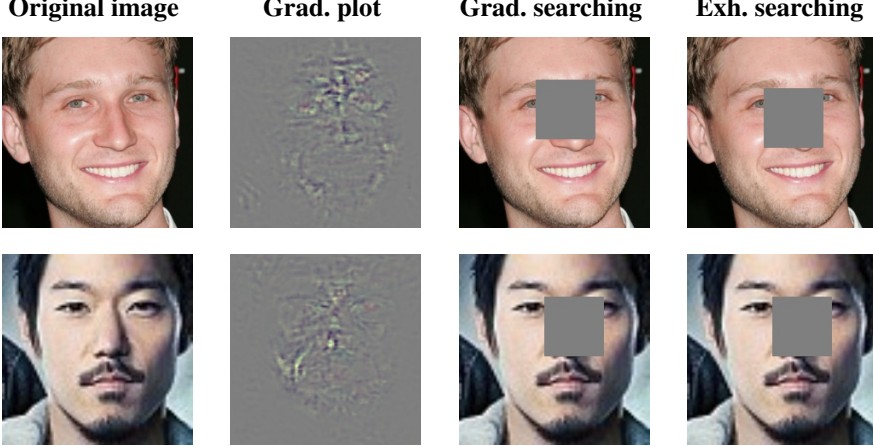

Figure 8: Examples of different search techniques. From left to right: 1) the original input image, 2) the plot of input gradient, 3) face with ROA location identified using gradient-based search, 4) face with ROA location identified using exhaustive search. Each row is a different example.

Figure 8 offers a visual illustration of how gradient-based search compares to exhaustive search for computing ROA.

## E  DETAILS OF PHYSICALLY REALIZABLE ATTACKS

Physically realizable attacks that we study have a common feature: first, they specify a mask, which is typically precomputed, and subsequently introduce adversarial noise inside the mask area. Let $M$ denote the mask matrix constraining the area of the perturbation $\delta$; $M$ has the same dimensions as the input image and contains 0s where no perturbation is allowed, and 1s in the area which can be perturbed. The physically realizable attacks we consider then solve an optimization problem of the following form:

$$\arg\max_{\boldsymbol{\delta}} L(f(\boldsymbol{x} + M\boldsymbol{\delta}; \boldsymbol{\theta}), y). \tag{4}$$

Next, we describe the details of the three physical attacks we consider in the main paper.

### E.1  EYEGLASS FRAME ATTACKS ON FACE RECOGNITION

Following Sharif et al. (2016), we first initialized the eyeglass frame with 5 different colors, and chose the best starting color by calculating the cross-entropy loss. For each update step, we divided the gradient value by its maximum value before multiplying by the learning rate which is 20. Then we only kept the gradient value of eyeglass frame area. Finally, we clipped and rounded the pixel value to keep it in the valid range.

### E.2  STICKER ATTACKS ON STOP SIGNS

Following Eykholt et al. (2018), we initialized the stickers on the stop signs with random noise. For each update step, we used the Adam optimizer with 0.1 learning rate and with default parameters. Just as for other attacks, adversarial perturbations were restricted to the mask area exogenously specified; in our case, we used the same mask as Eykholt et al. (2018)—a collection of small rectangles.

### E.3 ADVERSARIAL PATCH ATTACK

We used gradient ascent to maximize the log probability of the targeted class $P[y_{target}|x]$, as in the original paper (Brown et al., 2018). When implementing the adversarial patch, we used a square patch rather than the circular patch in the original paper; we don't anticipate this choice to be practically consequential. We randomly chose the position and direction of the patch, used the learning rate of 5, and fixed the number of attack iterations to 100 for each image. We varied the attack region (mask) $R \in \{0\%, 5\%, 10\%, 15\%, 20\%, 25\%\}$.

For the face recognition dataset, we used 27 images (9 classes (without targeted class) $\times$ 3 images in each class) to design the patch, and then ran the attack over 20 epochs. For the smaller traffic sign dataset, we used 15 images (15 classes (without targeted class) $\times$ 1 image in each class) to design the patch, and then ran the attack over 5 epochs. Note that when evaluating the adversarial patch, we used the validation set without the targeted class images.

## F ADDITIONAL EXPERIMENTS WITH DOA

### F.1 FACE RECOGNITION AND EYEGLASS FRAME ATTACK

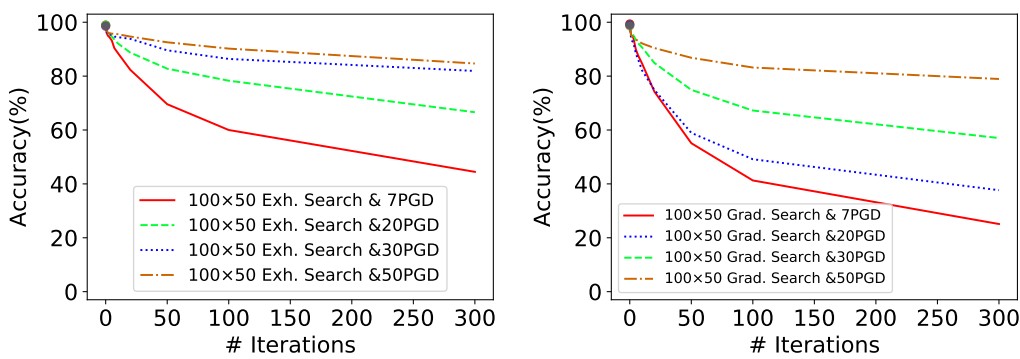

Figure 9: Effectiveness of DOA using the gradient-based method and the $100 \times 50$ region against the eyeglass frame attack, varying the number of PGD iterations for adversarial perturbations inside the rectangle. Left: using exhaustive search. Right: using gradient-based search.

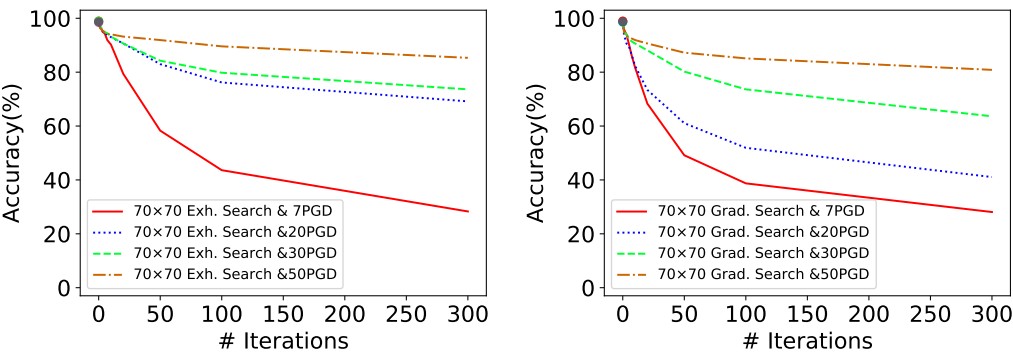

Figure 10: Effectiveness of DOA using the gradient-based method and the $70 \times 70$ region against the eyeglass frame attack, varying the number of PGD iterations for adversarial perturbations inside the rectangle. Left: using exhaustive search. Right: using gradient-based search.

## F.2 TRAFFIC SIGN CLASSIFICATION AND THE STOP SIGN ATTACK

Table 7: Comparison of effectiveness of different approaches against the stop sign attack. Best parameter choices were made for each.

|  | Validation Set | Attack Iterations | | | |
|---|---|---|---|---|---|
|  |  | 0 | $10^1$ | $10^2$ | $10^3$ |
| Clean Model | **98.69%** | **100%** | 42.5% | 32.5% | 25% |
| CAdv. Training: $\epsilon = 16$ | 96.95% | 97.5% | 57.5% | 45% | 42.5% |
| Randomized Smoothing: $\sigma = 0.25$ | 98.22% | **100%** | 70% | 64.5% | 62.5% |
| DOA (exhaustive; $7 \times 7$; 30 PGD) | 92.51% | **100%** | **92%** | **92%** | **90.5%** |
| DOA (gradient-based; $7 \times 7$; 30 PGD) | 95.82% | **100%** | 85% | 85.5% | 84% |

Table 8: Effectiveness of $10 \times 5$ DOA using exhaustive search with different numbers of PGD iterations against the stop sign attack.

|  | Validation Set | Attack Iterations | | | |
|---|---|---|---|---|---|
|  |  | 0 | $10^1$ | $10^2$ | $10^3$ |
| Clean Model | **98.69%** | **100%** | 42.5% | 32.5% | 25% |
| Exhaustive Search with 7 PGD | 96.50% | **100%** | 73% | 70.5% | 71% |
| Exhaustive Search with 20 PGD | 95.86% | **100%** | 86% | **85%** | **85%** |
| Exhaustive Search with 30 PGD | 95.59% | **100%** | **86.5%** | 84% | 82.5% |
| Exhaustive Search with 50 PGD | 95.87% | **100%** | 77% | 76.5% | 73.5% |

Table 9: Effectiveness of $10 \times 5$ gradient-based DOA with different numbers of PGD iterations against the stop sign attack.

|  | Validation Set | Attack Iterations | | | |
|---|---|---|---|---|---|
|  |  | 0 | $10^1$ | $10^2$ | $10^3$ |
| Clean Model | **98.69%** | **100%** | 42.5% | 32.5% | 25% |
| Gradient Based Search with 7 PGD | 97.53% | **100%** | **83.5%** | 78% | 78% |
| Gradient Based Search with 20 PGD | 97.11% | **100%** | 82.5% | 81.5% | 80.5% |
| Gradient Based Search with 30 PGD | 96.83% | **100%** | **83.5%** | 79.5% | 81% |
| Gradient Based Search with 50 PGD | 96.46% | **100%** | 82.5% | **82.5%** | **81.5%** |

Table 10: Effectiveness of $7 \times 7$ DOA using exhaustive search with different numbers of PGD iterations against the stop sign attack.

|  | Validation Set | Attack Iterations | | | |
|---|---|---|---|---|---|
|  |  | 0 | $10^1$ | $10^2$ | $10^3$ |
| Clean Model | **98.69%** | **100%** | 42.5% | 32.5% | 25% |
| Exhaustive Search with 7 PGD | 94.16% | **100%** | 86.5% | 85% | 85% |
| Exhaustive Search with 20 PGD | 92.74% | **100%** | 83% | 80% | 79% |
| Exhaustive Search with 30 PGD | 92.51% | **100%** | **92%** | **92%** | **90.5%** |
| Exhaustive Search with 50 PGD | 92.94% | **100%** | 89.5% | 89% | 88.5% |

Table 11: Effectiveness of $7 \times 7$ gradient-based DOA with different numbers of PGD iterations against the stop sign attack.

|  | Validation Set | Attack Iterations | | | |
|---|---|---|---|---|---|
|  |  | 0 | $10^1$ | $10^2$ | $10^3$ |
| Clean Model | **98.69%** | **100%** | 42.5% | 32.5% | 25% |
| Gradient Based Search with 7 PGD | 96.52% | **100%** | 81.5% | 80% | 79.5% |
| Gradient Based Search with 20 PGD | 95.51% | **100%** | 83.5% | 82.5% | 83% |
| Gradient Based Search with 30 PGD | 95.82% | **100%** | **85%** | **85.5%** | **84%** |
| Gradient Based Search with 50 PGD | 95.59% | **100%** | 81% | 81% | 81.5% |

## F.3 EVALUATION WITH THE ADVERSARIAL PATCH ATTACK

### F.3.1 FACE RECOGNITION

Table 12: Comparison of effectiveness of different approaches against the adversarial patch attack on the face recognition data. Best parameter choices were made for each.

|  | Size of Attacking Region | | | | | |
|---|---|---|---|---|---|---|
|  | 0% | 5% | 10% | 15% | 20% | 25% |
| Clean Model | 99.30% | 80.69% | 41.21% | 19.41% | 9.76% | 5.75% |
| CAdv. Training: $\epsilon = 16$ | 94.15% | 48.26% | 34.38% | 28.85% | 23.54% | 19.02% |
| Randomized Smoothing: $\sigma = 1$ | 98.83% | 61.99% | 36.26% | 33.80% | 24.68% | 16.02% |
| DOA (exh. ; $100 \times 50$; 50 PGD) | 99.30% | **98.72%** | **97.83%** | **97.18%** | **81.24%** | 24.19% |
| DOA (exh. ; $70 \times 70$; 50 PGD) | **99.41%** | 98.37% | 97.29% | 96.53% | 63.23% | **44.14%** |

Table 13: Effectiveness of adversarial training against the adversarial patch attack on face recognition data.

|  | Size of Attacking Region | | | | | |
|---|---|---|---|---|---|---|
|  | 0% | 5% | 10% | 15% | 20% | 25% |
| Clean Model | **99.30%** | **80.69%** | 41.21% | 19.41% | 9.76% | 5.75% |
| 7 Iterations Adv. Training: $\epsilon = 4$ | 92.28% | 57.05% | 39.15% | 34.71% | 10.52% | 7.81% |
| 7 Iterations Adv. Training: $\epsilon = 8$ | 66.54% | 27.98% | 23.54% | 21.26% | 18.55% | **17.03%** |
| 50 Iterations Adv. Training: $\epsilon = 4$ | 90.29% | 57.38% | **53.58%** | **36.01%** | **28.85%** | 16.05% |
| 50 Iterations Adv. Training: $\epsilon = 8$ | 53.68% | 21.80% | 21.15% | 17.79% | 17.79% | 16.70% |

Table 14: Effectiveness of curriculum adversarial training against the adversarial patch attack on face recognition data.

|  | Size of Attacking Region | | | | | |
|---|---|---|---|---|---|---|
|  | 0% | 5% | 10% | 15% | 20% | 25% |
| Clean Model | **99.30%** | 80.69% | 41.21% | 19.41% | 9.76% | 5.75% |
| CAdv. Training: $\epsilon = 4$ | 97.89% | **52.49%** | **45.44%** | 30.91% | 16.05% | 10.85% |
| CAdv. Training: $\epsilon = 8$ | 96.84% | 44.79% | 29.18% | 23.86% | 12.47% | 10.20% |
| CAdv. Training: $\epsilon = 16$ | 94.15% | 48.26% | 34.38% | 28.85% | 23.54% | **19.20%** |
| CAdv. Training: $\epsilon = 32$ | 81.05% | 42.95% | 40.46% | **32.00%** | **20.93%** | 17.14% |

Table 15: Effectiveness of randomized smoothing against the adversarial patch attack on face recognition data.

| | Size of Attacking Region | | | | | |
|---|---|---|---|---|---|---|
| | 0% | 5% | 10% | 15% | 20% | 25% |
| Clean Model | 99.30% | 80.69% | 41.21% | 19.41% | 9.76% | 5.75% |
| Randomized Smoothing: $\sigma = 0.25$ | 98.95% | 60.35% | 26.67% | 13.80% | 6.67% | 5.15% |
| Randomized Smoothing: $\sigma = 0.5$ | 97.66% | **81.05%** | **58.83%** | **33.92%** | 26.55% | 14.74% |
| Randomized Smoothing: $\sigma = 1$ | 98.83% | 61.99% | 36.26% | 33.80% | **24.68%** | **16.02%** |

Table 16: Effectiveness of DOA (50 PGD iterations) against the adversarial patch attack on face recognition data.

| | Size of Attacking Region | | | | | |
|---|---|---|---|---|---|---|
| | 0% | 5% | 10% | 15% | 20% | 25% |
| Clean Model | 99.30% | 80.69% | 41.21% | 19.41% | 9.76% | 5.75% |
| $(100 \times 50)$Exhaustive Search | 99.30% | **98.70%** | **97.83%** | **97.18%** | 81.24% | 24.19% |
| $(100 \times 50)$Gradient Based Search | **99.53%** | 98.41% | 96.75% | 87.42% | 57.48% | 24.62% |
| $(70 \times 70)$Exhaustive Search | 99.41% | 98.37% | 97.29% | 96.53% | 63.23% | **44.14%** |
| $(70 \times 70)$Gradient Based Search | 98.83% | 97.51% | 96.31% | 93.82% | **92.19%** | 31.34% |

## F.3.2 TRAFFIC SIGN CLASSIFICATION

Table 17: Comparison of effectiveness of different approaches against the adversarial patch attack on the traffic sign data. Best parameter choices were made for each.

| | Size of Attacking Region | | | | | |
|---|---|---|---|---|---|---|
| | 0% | 5% | 10% | 15% | 20% | 25% |
| Clean Model | **98.38%** | 75.87% | 58.62% | 40.68% | 33.10% | 22.47% |
| CAdv. Training: $\epsilon = 16$ | 96.77% | 86.93% | 70.21% | 66.11% | 51.13% | 40.77% |
| Randomized Smoothing: $\sigma = 0.5$ | 95.39% | 83.28% | 68.17% | 53.86% | 48.79% | 33.10% |
| DOA (grad. ; $7 \times 7$; 30 PGD) | 94.69% | **90.68%** | 88.41% | **81.79%** | **72.30%** | **58.71%** |
| DOA (grad. ; $7 \times 7$; 50 PGD) | 94.69% | 90.33% | **89.29%** | 78.92% | 70.21% | 58.54% |

Table 18: Effectiveness of adversarial training against the adversarial patch attack on traffic sign data.

| | Size of Attacking Region | | | | | |
|---|---|---|---|---|---|---|
| | 0% | 5% | 10% | 15% | 20% | 25% |
| Clean Model | 98.38% | 75.87% | 58.62% | 40.68% | 33.10% | 22.47% |
| Adv. Training: $\epsilon = 4$ | **98.96%** | **82.14%** | **64.46%** | 45.73% | 31.36% | 23.78% |
| Adv. Training: $\epsilon = 8$ | 95.62% | 76.74% | 62.89% | **46.60%** | **37.37%** | **25.09%** |

Table 19: Effectiveness of curriculum adversarial training against the adversarial patch attack on traffic sign data.

| | Size of Attacking Region | | | | | |
| --- | --- | --- | --- | --- | --- | --- |
| | 0% | 5% | 10% | 15% | 20% | 25% |
| Clean Model | 98.38% | 75.87% | 58.62% | 40.68% | 33.10% | 22.47% |
| CAdv. Training: $\epsilon = 4$ | **98.96%** | 77.18% | 57.32% | 39.02% | 38.94% | 31.10% |
| CAdv. Training: $\epsilon = 8$ | **98.96%** | 86.24% | 68.64% | 60.19% | 42.42% | 35.10% |
| CAdv. Training: $\epsilon = 16$ | 96.77% | **86.93%** | **70.21%** | **66.11%** | **51.13%** | **40.77%** |
| CAdv. Training: $\epsilon = 32$ | 63.78% | 59.93% | 51.57% | 45.38% | 37.37% | 27.79% |

Table 20: Effectiveness of randomized smoothing against the adversarial patch attack on traffic sign data.

| | Size of Attacking Region | | | | | |
| --- | --- | --- | --- | --- | --- | --- |
| | 0% | 5% | 10% | 15% | 20% | 25% |
| Clean Model | **98.38%** | 75.87% | 58.62% | 40.68% | 33.10% | 22.47% |
| Randomized Smoothing: $\sigma = 0.25$ | 98.27% | 82.24% | 66.78% | **54.67%** | 40.37% | **33.79%** |
| Randomized Smoothing: $\sigma = 0.5$ | 95.39% | **83.28%** | **68.17%** | 53.86% | **48.79%** | 33.10% |
| Randomized Smoothing: $\sigma = 1$ | 85.47% | 74.39% | 54.44% | 45.44% | 40.48% | 29.06% |

Table 21: Effectiveness of DOA (30 PGD iterations) against the adversarial patch attack on traffic sign data.

| | Size of Attacking Region | | | | | |
| --- | --- | --- | --- | --- | --- | --- |
| | 0% | 5% | 10% | 15% | 20% | 25% |
| Clean Model | **98.38%** | 75.87% | 58.62% | 40.68% | 33.10% | 22.47% |
| $(10 \times 5)$Exhaustive Search | 94.00% | **91.46%** | 86.67% | 76.39% | 68.12% | 55.57% |
| $(10 \times 5)$Gradient Based Search | 93.77% | 86.67% | 80.57% | 75.00% | 64.90% | 54.44% |
| $(7 \times 7)$Exhaustive Search | 92.27% | 89.02% | 82.93% | 78.40% | 66.90% | 55.92% |
| $(7 \times 7)$Gradient Based Search | 94.69% | 90.68% | **88.41%** | **81.79%** | **72.30%** | **58.71%** |

Table 22: Effectiveness of DOA (50 PGD iterations) against the adversarial patch attack on traffic sign data.

| | Size of Attacking Region | | | | | |
| --- | --- | --- | --- | --- | --- | --- |
| | 0% | 5% | 10% | 15% | 20% | 25% |
| Clean Model | **98.38%** | 75.87% | 58.62% | 40.68% | 33.10% | 22.47% |
| $(10 \times 5)$Exhaustive Search | 96.42% | **93.90%** | 90.42% | 80.66% | 67.77% | 53.92% |
| $(10 \times 5)$Gradient Based Search | 93.54% | 90.33% | 85.80% | 79.79% | 69.86% | 49.74% |
| $(7 \times 7)$Exhaustive Search | 87.08% | 82.49% | 75.96% | 70.99% | 60.10% | 52.53% |
| $(7 \times 7)$Gradient Based Search | 94.69% | 90.33% | **89.29%** | 78.92% | 70.21% | **58.54%** |

## G  EXAMPLES OF PHYSICALLY REALIZABLE ATTACK AGAINST ALL DEFENSE MODELS

### G.1  FACE RECOGNITION

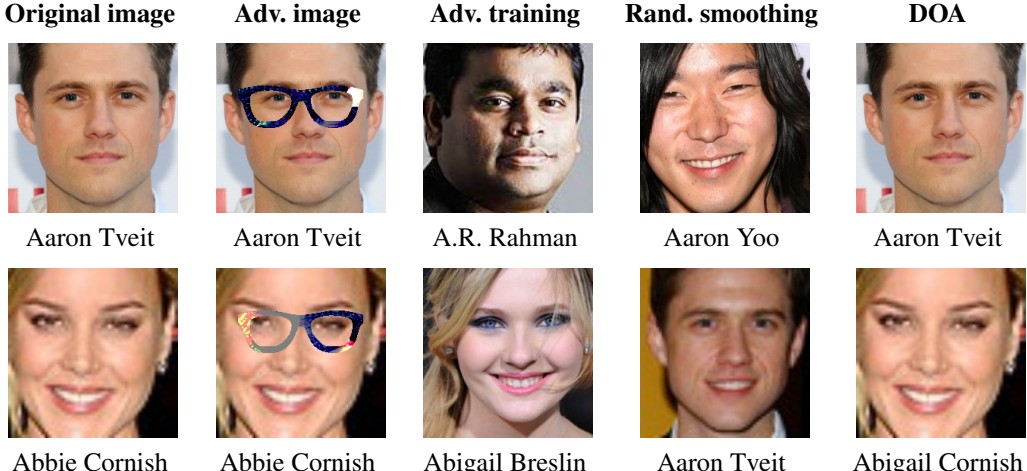

Figure 11: Examples of the eyeglass attack on face recognition. From left to right: 1) the original input image, 2) image with adversarial eyeglass frames, 3) face predicted by a model generated through adversarial training, 4) face predicted by a model generated through randomized smoothing, 5) face predicted (correctly) by a model generated through DOA. Each row is a separate example.

### G.2  TRAFFIC SIGN CLASSIFICATION

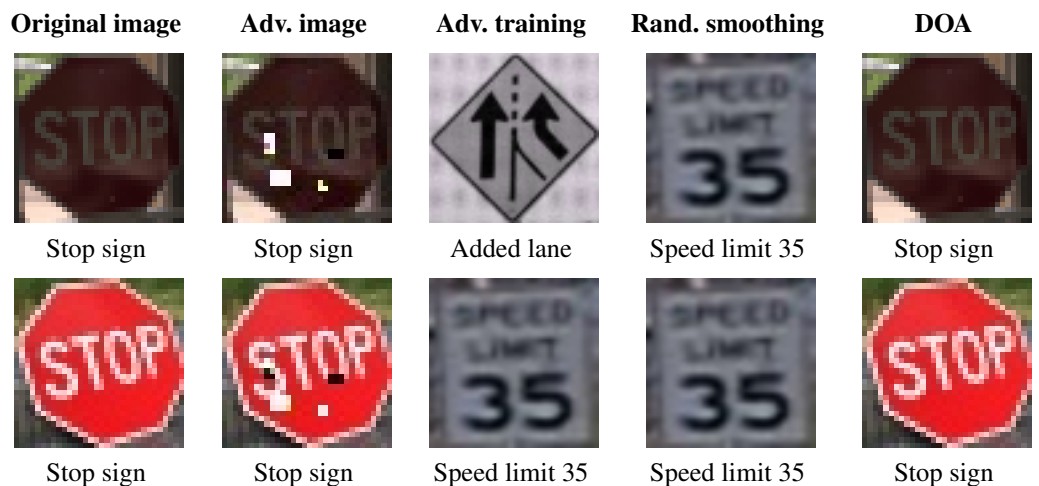

Figure 12:  Examples of the stop sign attack. From left to right: 1) the original input image, 2) image with adversarial eyeglass frames, 3) face predicted by a model generated through adversarial training, 4) face predicted by a model generated through randomized smoothing, 5) face predicted (correctly) by a model generated through DOA. Each row is a separate example.

## H    EFFECTIVENESS OF DOA METHODS AGAINST $l_\infty$ ATTACKS

For completeness, this section includes evaluation of DOA in the context of $l_\infty$-bounded attacks implemented using PGD, though these are outside the scope of our threat model.

Table 23: Effectiveness of DOA (50 PGD iterations) against 20 Iterations $L_\infty$ Attacks on Face Recognition

|  | Attack Strength | | | | |
| --- | --- | --- | --- | --- | --- |
|  | $\epsilon = 0$ | $\epsilon = 2$ | $\epsilon = 4$ | $\epsilon = 8$ | $\epsilon = 16$ |
| Clean Model | 98.94% | 44.04% | 1.70% | 0% | 0% |
| $(100 \times 50)$Exhaustive Search | 98.72% | 39.79% | 1.06% | 0% | 0% |
| $(100 \times 50)$Gradient Based Search | 98.51% | 40.64% | 3.40% | 0% | 0% |
| $(70 \times 70)$Exhaustive Search | 98.51% | 35.74% | 0.43% | 0% | 0% |
| $(70 \times 70)$Gradient Based Search | 97.45% | 33.40% | 0.43% | 0% | 0% |

Table 24: Effectiveness of DOA (50 PGD iterations) against 20 Iterations $L_\infty$ Attacks on Traffic Sign Classification

|  | Attack Strength | | | | |
| --- | --- | --- | --- | --- | --- |
|  | $\epsilon = 0$ | $\epsilon = 2$ | $\epsilon = 4$ | $\epsilon = 8$ | $\epsilon = 16$ |
| Clean Model | 98.69% | 89.54% | 61.58% | 24.65% | 5.14% |
| $(10 \times 5)$Exhaustive Search | 95.87% | 91.55% | 76.57% | 39.02% | 7.75% |
| $(10 \times 5)$Gradient Based Search | 96.46% | 91.81% | 78.83% | 46.86% | 7.84% |
| $(7 \times 7)$Exhaustive Search | 92.94% | 89.54% | 77.09% | 42.77% | 5.83% |
| $(7 \times 7)$Gradient Based Search | 95.59% | 91.20% | 79.52% | 46.86% | 6.70% |

Table 23 presents results of several variants of DOA in the context of PGD attacks in the context of face recognition, while Table 24 considers these in traffic sign classification. The results are quite consistent with intuition: DOA is largely unhelpful against these attacks. The reason is that DOA fundamentally assumes that the attacker only modifies a relatively small proportion ($\sim$5%) of the scene (and the resulting image), as otherwise the physical attack would be highly suspicious. $l_\infty$ bounded attacks, on the other hand, modify all pixels.

# I EFFECTIVENESS OF DOA METHODS AGAINST $l_0$ ATTACKS

In addition to considering physical attacks, we evaluate the effectiveness of DOA against Jacobian-based saliency map attacks (JSMA) (Papernot et al., 2015) for implementing $l_0$-constraint adversarial examples. As Figure 13 shows, in both the face recognition and traffic sign classification, DOA is able to improve classificatio robustness compared to the original model.

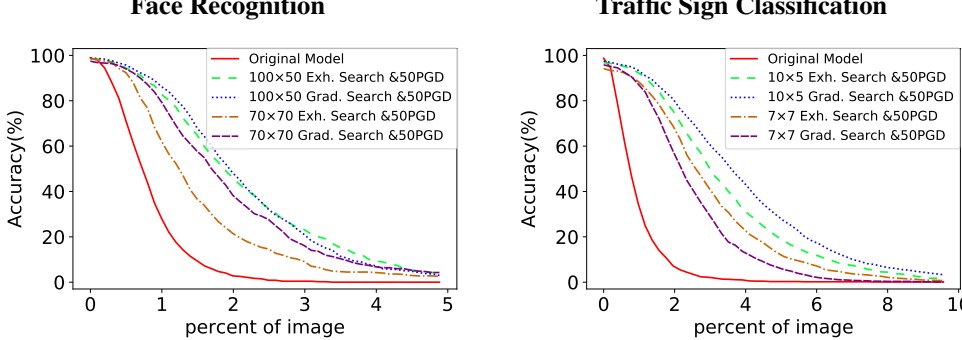

Figure 13: $L_0$ attacks on face recognition and traffic sign classification.

## J    EFFECTIVENESS OF DOA AGAINST OTHER MASK-BASED ATTACKS

To further illustrate the ability of DOA to generalize, we evaluate its effectiveness in the context of three additional occlusion patterns: a union of triangles and circle, a single larger triangle, and a heart pattern.

As the results in Figures 14 and 15 suggest, DOA is able to generalize successfully to a variety of physical attack patterns. It is particularly noteworthy that the larger patterns (large triangle—middle of the figure, and large heart—right of the figure) are actually quite suspicious (particularly the heart pattern), as they occupy a significant fraction of the image (the heart mask, for example, accounts for 8% of the face).

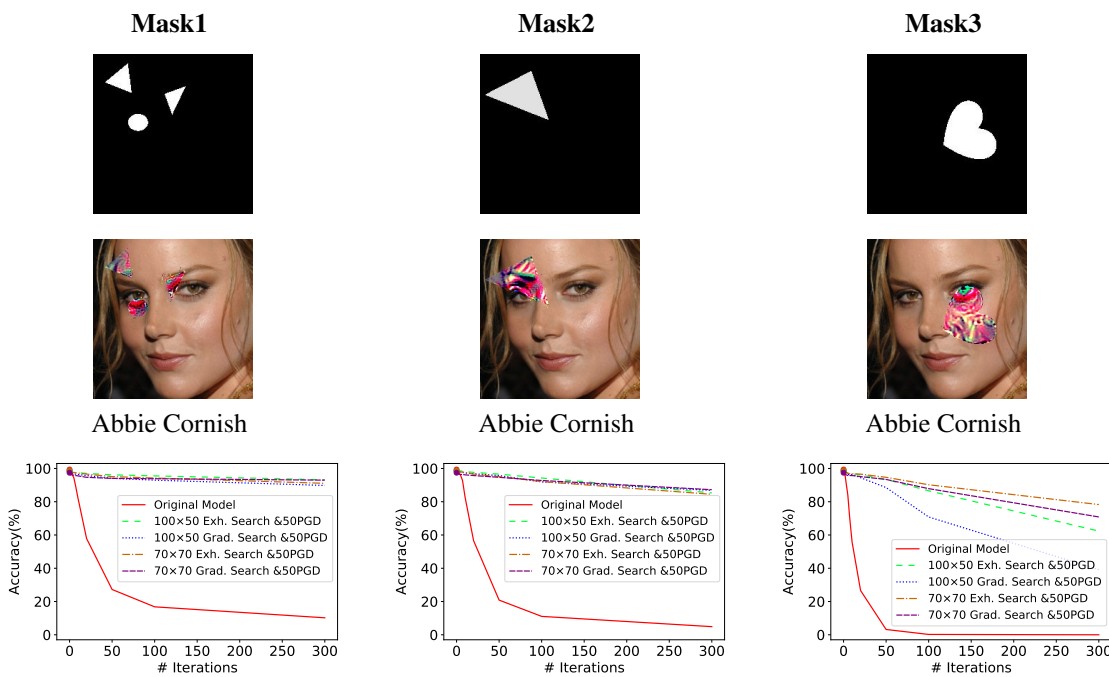

Figure 14: Additional mask-based attacks on face recognition.

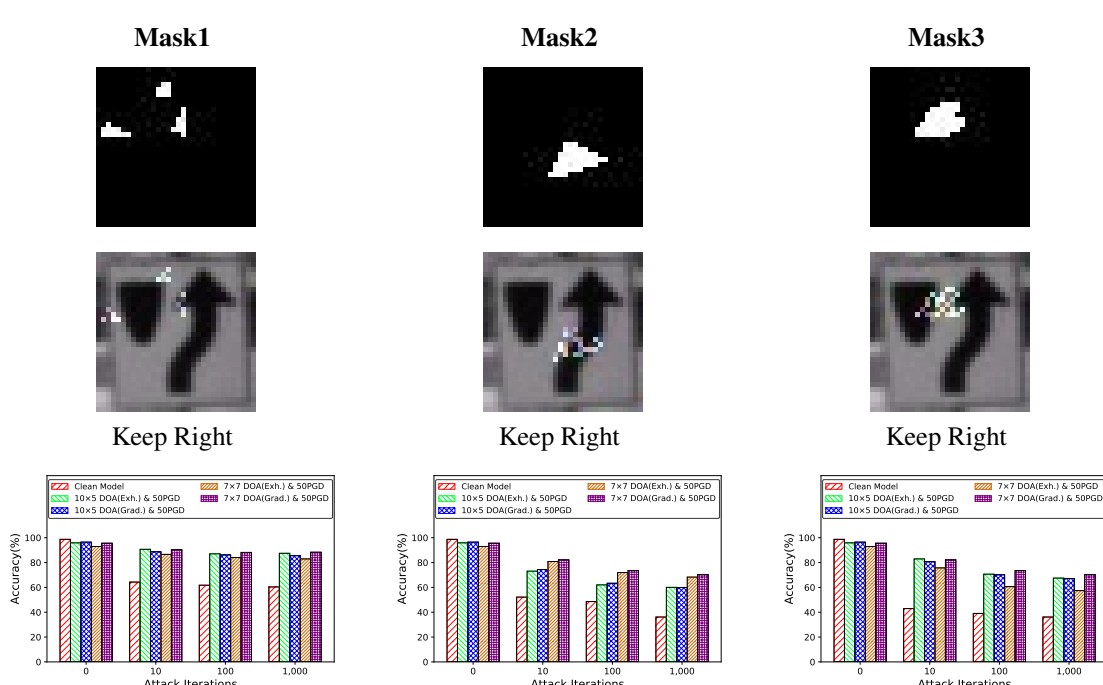

Figure 15: Additional mask-based attacks on traffic sign classification.

