# OpenReview forum: "Defending Against Physically Realizable Attacks on Image Classification"
_ICLR.cc/2020/Conference — Accept (Spotlight)_

### Official Review · AnonReviewer1 · 2019-10-22
**Official Blind Review #1**

**Rating:** 8

**Review:**

The paper aims to provide a defense to physically realizable attacks for image classifiers. First, the paper demonstrates that Lp-ball robustness (obtained via adversarial training or randomized smoothing) do not necessarily result in robustness to physical attacks such as adversarial stickers. Next, the paper proposes a variant of adversarial training (using adversarial rectangles) and shows empirically that such a method results in much improved robustness to the aforementioned physical attacks.

I vote to accept this paper. It has clear motivation, is very clearly written, evaluates proper benchmarks from recent literature, and proposes a new method that shows a clear improvement over benchmarks in literature. The goal is clearly laid out in Section 3, and the paper justifies its claims via various experiments. Overall, I think it is a good contribution to the adversarial examples literature, as it provides robustness against more “real-world” attacks.

In particular, the results from Fig. 2 and Fig. 3, while not surprising, appear to be done thoroughly, as the authors evaluate against various forms of adversarial training and various degrees of randomized smoothing. Later, the results in Fig. 4 and Fig. 5 show a clear benefit from their method.

I do have a few pieces of feedback that I think could improve the work.

In section 5.1, do you really only train for 5 epochs? Is this just a fine-tuning procedure after standard training is performed, or is the entire training procedure 5 epochs? That seems especially short for getting any amount of accuracy. I think it is worth clarifying.

In Section 3 (or in the Related Works), I would suggest that the authors do mention related works about other types of realistic adversarial examples or perturbations, such as those generated via physical transformations like translations and rotations [1] or even common corruption robustness [2]. This is especially relevant since the authors claim to be the “first investigation of this issue in computer vision applications,” so it’s worth clarifying that the authors do not claim to be the first work about robustness to all realistic perturbations.

One thing I would be curious to see is if the DOA training method provides any robustness to standard Lp-ball adversarial examples or to perturbations like rotations.

Additional Feedback:

- In Related Works, I think there are some places where it would look better to use parentheses around the citations.

[1] https://arxiv.org/abs/1712.02779
[2] https://arxiv.org/abs/1903.12261

**Experience Assessment:**

I have published one or two papers in this area.

**Review Assessment: Checking Correctness Of Derivations And Theory:**

N/A

**Review Assessment: Checking Correctness Of Experiments:**

I assessed the sensibility of the experiments.

**Review Assessment: Thoroughness In Paper Reading:**

I made a quick assessment of this paper.

---

> ### Author Response · Authors · 2019-11-08
> **Author Response to Review #1**
>
> We thank the reviewer for the thoughtful comments and suggestions.
>
> 1) Comment: "In section 5.1, do you really only train for 5 epochs? Is this just a fine-tuning procedure after standard training is performed, or is the entire training procedure 5 epochs? That seems especially short for getting any amount of accuracy. I think it is worth clarifying."
>
> Response:
>
> This is just a fine-tuning procedure after standard training is performed.
>
> 2) Comment: adding suggested related work, and clarifying our claims.
>
> Response:
>
> We have done as the reviewer suggested; please see the updated draft.
>
> 3) Comment: "One thing I would be curious to see is if the DOA training method provides any robustness to standard Lp-ball adversarial examples or to perturbations like rotations."
>
> Response:
>
> We added experimental results with l_infty attacks.  As we would expect, DOA doesn't help against these attacks.

---

> > ### Comment · AnonReviewer1 · 2019-11-12
> > **Thanks**
> >
> > Thank you for addressing my remaining comments. I will keep my score the same.

---

### Official Review · AnonReviewer2 · 2019-10-25
**Official Blind Review #2**

**Rating:** 8

**Review:**

[Note: I gave a 3 for thoroughness even though I only read the paper once, because I believe that I carefully considered the paper while reading it.]

This paper argues that threat models such as L-inf are limited when considering physically realizable attacks, and provides evidence for this by showing that L-inf adversarial training is insufficient to fully confer robustness against physically realizable attacks in the literature such as the adversarial glasses attack. The paper then proposes an alternate threat model based on contiguous rectangular regions, and shows that adversarial training against this model does far better.

Overall this is a strong paper with thorough experiments, and was for the most part carefully written (although see some quibbles below). I particular liked the paper's carefully distinction between physically *realizable* and physically *realized* attacks, and the admission that not all realizable attacks would fall under the threat model (while still justifying why the model is interesting).

I am on the border of weak accept and strong accept. The main two points keeping me from strong accept are discussed below (and seem perhaps addressable by the authors):

1. The assertion that rectangular occlusions might be a fruitful model for realizable attacks is only lightly tested, since the attacks considered in this paper all fall into the rectangular occlusions threat model. Since one of the key points in the paper is that training in the L-inf threat model could leave vulnerabilities to non-Linf attacks, we might expect the same with rectangular occlusions. A more convincing demonstration would be to test robustness to physically realizable attacks that fall outside the rectangular model (perhaps even skewed or rotated rectangles, although a less synthetic example would be even better). A more minor point is that it would be nice to test the robustness of models trained on rectangular occlusions against L-inf adversarial attacks. This is relevant to knowing whether training on occlusions is giving up on adversarial robustness or actually helps against both (I actually think it's plausible that you would do well on L-inf, but it seems worth testing either way).

2. The high-level claimed take-away that adversarial training does not help against physically realizable attacks seems false in light of Figure 2, which shows that L-inf adversarial training substantially improves robustness relative to the baseline. A more accurate take-away would be that on some datasets adversarial training helps but still leaves a gap, while on others it does not help at all and perhaps hurts. I would prefer more careful wording as a reader who only goes through the introduction might not see Figure 2.

On #1, I should stress that the paper would still be interesting regardless of the outcome of the two experiments in #1; I just think that for thoroughness it would be nice to include them. The existing experiments are already thorough so this would be going above and beyond, but that is why it might help raise my score from weak to strong accept.

EDITED TO ADD: For #2, I would also be happy with an argument from the authors as to why the current language appropriately describes the experiments. I do not wish to dictate to the authors what their take-aways are, but more to open up for discussion a point that seemed slightly sloppy to me.

Minor comments:

Please avoid subjective intensifiers: "We then use an extensive experimental evaluation to demonstrate that our proposed approach is far more robust against physical attacks on deep neural networks than adversarial training and randomized smoothing methods that leverage lp-based attack models." Both "extensive" and "far" are unnecessary.

Make sure to use \citep vs \citet correctly.

First sentence of 2.1 is too verbose. Overall the prose in that paragraph is turgid, due to too many action phrases being turned into nouns. E.g. "The focus is on", "The typical goal is" both indicate *action* and could be profitably turned into verbs. See Williams and Bizup's book on Style.

"since our ultimate goal is to defend against physical attacks, untargeted attacks that aim to maximize error are the most useful" This seems weak; I don't understand why an attack being physical should go in line with it being untargeted; oftentimes an attacker will have a specific targeted goal.

"another advantage of the ROA attack is that it is, in principle,easier to compute than,  say,l∞-bounded attacks" This seems incongruous with the subsequent text, which admits that ROA if implemented naively would be slower than L-inf attacks.

**Experience Assessment:**

I have published in this field for several years.

**Review Assessment: Checking Correctness Of Derivations And Theory:**

N/A

**Review Assessment: Checking Correctness Of Experiments:**

I assessed the sensibility of the experiments.

**Review Assessment: Thoroughness In Paper Reading:**

I read the paper at least twice and used my best judgement in assessing the paper.

---

> ### Author Response · Authors · 2019-11-08
> **Author Response to Review #2**
>
> We appreciate the thoughtful comments and suggestions by the reviewer.
>
> 1) Comment: "The assertion that rectangular occlusions might be a fruitful model for realizable attacks is only lightly tested, since the attacks considered in this paper all fall into the rectangular occlusions threat model. [...] A more convincing demonstration would be to test robustness to physically realizable attacks that fall outside the rectangular model (perhaps even skewed or rotated rectangles, although a less synthetic example would be even better)."
>
> Response:
>
> We appreciate this point; indeed, we tried to be as comprehensive on this score as we could given existing attacks.  Below, we describe why the experiments in the submitted draft already fulfill the criterion of "testing robustness to physically realizable attacks that fall outside the rectangular model", as well as additional experiments we conducted in response to the comment to further bolster this claim.
>
> a) Existing experiments: we evaluated DOA using 3 attacks.  The first involves adversarially designed eyeglass frames, which are non-rectangular (contrast Fig. 1a with Fig. 6).  The second involves adversarial stickers on stop signs; this one has several disjoint stickers, which is quite unlike our single-rectangle attack (contrast Fig. 1b with Fig. 7, noting the contiguity of the region in Fig. 7).  Only our third attack involving adversarial patches is qualitatively similar to our ROA model.
>
> b) We ran several more experiments with a non-rectangular adversarial patches (masks using a disjoint mixture of triangles and a circle, a triangle, and a heart); these results are largely consistent with the rest, and have been added in Appendix I; please see the uploaded revised draft.
>
> 2) Comment: "A more minor point is that it would be nice to test the robustness of models trained on rectangular occlusions against L-inf adversarial attacks."
>
> Response:
>
> We did this.  As expected, DOA does not perform well against L-inf attacks.  The new results have been added in Appendix H of the revised draft.
>
> 3) Comment: "The high-level claimed take-away that adversarial training does not help against physically realizable attacks seems false in light of Figure 2, which shows that L-inf adversarial training substantially improves robustness relative to the baseline. A more accurate take-away would be that on some datasets adversarial training helps but still leaves a gap, while on others it does not help at all and perhaps hurts. I would prefer more careful wording as a reader who only goes through the introduction might not see Figure 2."
>
> Response:
>
> We agree with the reviewer that our statement was imprecise.  We amended the language in the introduction to "we study the performance on adversarial training and randomized smoothing against these attacks, and show that both have limited effectiveness in this context (quite ineffective in some settings, and somewhat more effective, but still not highly robust, in others)", following the reviewer's suggestion (please see the updated draft of the paper).
>
> 4) Comment: "avoid subjective intensifiers"
>
> Response: we removed the "extensive" intensifier; however, we feel that the second case (which we replaced by "significant") is warranted to communicate that the comparison is in most cases not close.
>
> 5) Comment: "\citet vs. \citep"
>
> Response: we believe we fixed all the issues.
>
> 6) Comment: Poorly written first paragraph of 2.1.
>
> Response: we rewrote this paragraph; please see the revised draft.
>
> 7) Comment: "'since our ultimate goal is to defend against physical attacks, untargeted attacks that aim to maximize error are the most useful' This seems weak; I don't understand why an attack being physical should go in line with it being untargeted; oftentimes an attacker will have a specific targeted goal."
>
> Response: what we meant was that untargeted attacks are stronger, and since have no prior knowledge of the possible attack target, the most practical approach for defense is to assume that the attack is untargeted.  We rewrote this sentence as follows: "An important feature of this attack is that it is untargeted: since our ultimate goal is to defend against physical attacks whatever their target, considering untargeted attacks obviates the need to have precise knowledge about the attacker's goals." Please see the revised draft.
>
> 8) Comment: "'another advantage of the ROA attack is that it is, in principle, easier to compute than,  say,l∞-bounded attacks' This seems incongruous with the subsequent text, which admits that ROA if implemented naively would be slower than L-inf attacks."
>
> Response: we agree; we rewrote this sentence.

---

> > ### Comment · AnonReviewer2 · 2019-11-12
> > **Thanks!**
> >
> > Thank you for your detailed reply. This satisfactorily addresses my points and I have updated my score accordingly.
> >
> > A minor quibble regarding eyeglasses is that although they are not rectangular, they are a subset of a rectangle that is not *too* much bigger than the rectangles used at training time, so are not that far from falling within the threat model. On the other hand I am satisfied that overall the considered attacks do a reasonable job at stress-testing the threat model. It would be interesting to eventually consider even more comprehensive tests but probably unreasonable to expect that in the current submission.

---

### Official Review · AnonReviewer4 · 2019-10-28
**Official Blind Review #4**

**Rating:** 3

**Review:**

First of all, the two physical attacks evaluated in this paper have similar attacking patterns, i.e., mask-based pixel attacks. So it is not surprising that DOA is more robust in these cases since DOA is trained on this attacking pattern.

Actually it has been shown that the framework of adversarial training (AT) will overfit to the attacking patterns used in training. For example, PGD-AT models are less robust to simple non-pixel transformation, like rotation, than the standard models [1]. So what DOA does is just substituting the PGD module in AT to overfit the new attacking patterns, which is of limited contribution and novelty.

Besides, AT is not really scalable compared to other simpler defense strategies like input transformation. [2] proposes a simple and effective defense based on different combinations of input transformation and its performance even surpasses some SOTA AT models with less computation.

Another advantage of these off-the-shelf defenses like input transformation is that they do not depend on the specific details of attacks, so they are more reliable when you are unknown about the potential attacking patterns in practice. In comparison, there is an implicit assumption in AT methods that the attacking patterns in training and test are similar. This is the reason why PGD-AT is not robust facing mask-based physical attacks or simple rotations.

So under the more completed and flexible physical attacks, a defense based on the AT framework like DOA may not be a good choice. Although AT methods are quite effective and widely studied under the l_p attacks, the authors are expected to consider more factors if they really want to design a robust system in the physical world, rather than just follow or apply the most popular pipeline like AT.

Reference:
[1] Engstrom et al. A rotation and a translation suffice: Fooling cnns with simple transformations. ICML 2019
[2] Raff et al. Barrage of Random Transforms for Adversarially Robust Defense. CVPR 2019

**Experience Assessment:**

I have published in this field for several years.

**Review Assessment: Checking Correctness Of Derivations And Theory:**

N/A

**Review Assessment: Checking Correctness Of Experiments:**

I carefully checked the experiments.

**Review Assessment: Thoroughness In Paper Reading:**

I read the paper thoroughly.

---

> ### Author Response · Authors · 2019-11-08
> **Author Response to Review #4 (part 2)**
>
> 3) Comment: "AT is not really scalable compared to other simpler defense strategies like input transformation"
>
> Response:
>
> We suspect that the reviewer means that AT(PGD) is not scalable.  However, scalability of AT(X(f)) in general depends on the nature of X(f).  For example, standard data augmentation can be viewed as constant X (i.e., independent of model f), and scalability there is not an issue.  What we demonstrate is that DOA is scalable, essentially because the ROA attack model has far fewer degrees of freedom than typical lp-bounded attacks (such as PGD).
>
> 4) Comment: "Another advantage of these off-the-shelf defenses like input transformation is that they do not depend on the specific details of attacks, so they are more reliable when you are unknown about the potential attacking patterns in practice."
>
> Response:
>
> We would argue that what ultimately matters is the generalizability of a defense with respect to the target (and broad) threat model.  Our central argument is that AT(ROA) (what we call DOA) in fact generalizes to physical masking/occlusion attacks using very different patterns (eyeglass frames, graffiti-like patterns on stop signs).  This, in turn, demonstrates that DOA does not in fact rely on specific knowledge about attack patterns, but yields generalized robustness.  Our threat model is, however, restricted to masking attacks; consequently, we would not expect DOA to be effective against, say, l_p-bounded attacks that can directly modify all of the pixels in the image.
>
> 5) Comment: "In comparison, there is an implicit assumption in AT methods that the attacking patterns in training and test are similar."
>
> Response:
>
> Our central argument is in fact that AT(ROA) yields general robustness to physical attack patterns that are quite different from ROA (eyeglass frames, graffiti-like patterns on stop signs).  This is the primary purpose of Section 5 of the paper (see also additional new data in Appendix I).  In other words, AT(ROA) is robust to attacks that use very different patterns from those used in training.  A conceptually similar argument in the PDF malware detection domain was made by [1], who also exhibit an attack X(f) such that AT(X(f)) is robust to attacks that are completely different from X(f).
>
> Our larger point is to contest the conventional thinking in AML that AT(X(f)) requires attacks at test time to be similar to X(f).  We provide additional evidence that what ultimately matters is what model X(f) is used by adversarial training.
>
> [1] Tong, et al. Improving robustness of ML classifiers against realizable evasion attacks using conserved features.  USENIX Security, 2019.

---

> > ### Comment · AnonReviewer4 · 2019-11-09
> > **Thank you for the response**
> >
> > Thank you for the author's response, but some of the answers make me more confused about the proposed method DOA=AT(ROA).
> >
> > My original opinion is that ROA is a high-level abstract of the attacking patterns in the evaluated physical attacks, which is at least reasonable as a motivation for the proposed method. However, in the rebuttal, the authors repeatedly claim that ROA is a very different pattern from the considered physically realizable attacks. Then I wonder what is the motivation and justification to use ROA in AT to defend these physically realizable attacks, if the authors insist on the attacking patterns between them are very different?
> >
> > I read the Tong et al. 2019 paper provided by the authors, and I find no theoretical analysis to support the conclusion about AT(X(f)), so I think it is just an empirical observation. Actually  AT(PGD) is also robust to many different attacks, e.g., JSMA ($l_{0}$), CW($l_{2}$), SPSA, ZOO and more query-based black-box attacks. Since PGD is under $l_{\infty}$ pattern, we would consider AT(PGD) can well generalize to other attack patterns before we found that it is not robust to rotation (Engstrom et al. 2019).
> >
> > Besides, if AT(ROA) is more generalizable than AT(PGD), I would like to know if AT(ROA) is as robust as AT(PGD) on defending the pixel-based PGD attack in the digital world. If yes, I think this work is much more significant than only applied to physical attacks, since it can substitute the most commonly used PGD in AT and accelerate the training process with a simpler ROA. But if not, that would be overfitting in AT(ROA), just like AT(PGD).

---

> > > ### Author Response · Authors · 2019-11-10
> > > **Re: Thank you for the response**
> > >
> > > We genuinely appreciate the reviewer's efforts to understand our work, as well as spending time to read the referenced paper (this, indeed, goes above and beyond!)
> > >
> > > The issue appears (to us) to be largely about interpreting our claims in both the paper and the response.  Let us attempt to clarify:
> > >
> > > 1) Comment: "My original opinion is that ROA is a high-level abstract of the attacking patterns in the evaluated physical attacks"
> > >
> > > Response: this is exactly correct!  What we wrote in response is in no way meant to contradict this.  We attempt to explain why our response is fully consistent with this next.
> > >
> > > 2) Comment: "in the rebuttal, the authors repeatedly claim that ROA is a very different pattern from the considered physically realizable attacks"
> > >
> > > Response: It seems that we agree that an abstraction is, by its very nature, different from the thing being abstracted.  In our case, we want a simple, generic, representation of masking attacks that does not rely on a *particular* pattern involved (which, naturally, varies by context; e.g., face recognition attacks involve different patterns than stop sign attacks).  This is the role of ROA---a generic, abstract, representation of occlusion attacks.  When we say "different", we mean precisely this: in our evaluation, we validate that the abstraction does reasonably abstract the attacks using very different patterns, so long as these still involve masking.
> > >
> > > 3) Comment: "Then I wonder what is the motivation and justification to use ROA in AT to defend these physically realizable attacks, if the authors insist on the attacking patterns between them are very different?"
> > >
> > > Response: we hope our response above already addresses it.  And while we can make intuitive arguments about why this abstraction is reasonable (it involves generic masking, etc), ultimately the most convincing evidence is what we present in Section 5: it works!
> > >
> > > 4) Comment: "I read the Tong et al. 2019 paper provided by the authors, and I find no theoretical analysis to support the conclusion about AT(X(f)), so I think it is just an empirical observation"
> > >
> > > Response: correct, it is an empirical observation, but the evidence seems to us fairly compelling in that domain (granted, it's a subjective point).  In our view, this is in fact what's most important: we can present many models and prove many theorems, but what ultimately matters is if things actually work in practice.  We can only verify this through empirical observations.  This is not to disparage theory; rather, both theoretical and empirical work have fundamental roles in science.
> > >
> > > 5) Comment: "we would consider AT(PGD) can well generalize to other attack patterns before we found that it is not robust to rotation"
> > >
> > > Response: we are entirely agnostic on this point.  In general, it seems a useful endeavor to understand the scope of generalizability of any defense; we don't believe there exists any approach that defeats every attack.
> > >
> > > Actually, let us suggest what we feel is a useful framework for thinking about this sort of thing.  Suppose we have an abstract attack model, X(f).  Let Y(f) we a target space of attacks, with X(f) ne Y(f).   We can say that X(f) generalizes to Y(f) if AT(X(f)) defends against "most" attacks in Y(f).
> > >
> > > So, in our case, Y(f) are physical occlusion attacks (with different possible masks).  Perhaps for PGD is PGD U CW U ... (whatever other attacks it can defend against).
> > >
> > > In other words, generalizability is conditional (it's a property of what your ultimate target threat space is).
> > >
> > > 6) Comment: "AT(ROA) is more generalizable than AT(PGD), I would like to know if AT(ROA) is as robust as AT(PGD) on defending the pixel-based PGD attack in the digital world."
> > >
> > > Response: we absolutely do not claim this.  In fact, evidence suggests it does not.
> > >
> > > 7) Comment: "But if not, that would be overfitting in AT(ROA), just like AT(PGD)"
> > >
> > > Response: this seems to view advances as all-or-nothing: either we defeat all possible attacks, or we have made no progress.  We disagree with this perspective.  We address a particular class of attacks, which no other defense succeeds against.  We argue that this class is important (the reviewer does not seem to contest this).  Ergo, we have made a clear advance on the state of the art.   We do not understand why failing to defeat attacks that are not in the scope of our threat space qualifies as overfitting.  By this definition, every security paper overfits to an attack.  Indeed, by this definition, every defense against adversarial examples ever published overfits to the attacks it defends against.

---

> ### Author Response · Authors · 2019-11-08
> **Author Response to Review #4 (part 1)**
>
> We appreciate the reviewer's comments.
>
> Before proceeding with a detailed response, we wish to make a meta-point that we hope will facilitate subsequent discussion.
>
> Specifically, we view AT (adversarial training) as a function AT(X(f)), where X(f) is an attack model (or, more generically, some adversarial loss function that takes the DNN model f as input, such as convex-relaxation and duality-based approaches for certified robust training).  X(f) depends on f, in general, because in adversarial example (or "test-time", "decision-time", "inference-time", etc.) attacks, the attack optimizes in response to the given model f.
>
> This perspective reduces defense, or robustness, to the issue of identifying the right X(f).  There is recent evidence that this general conceptual approach makes sense [1] (we'll say more about this below), in contrast to the conventional perspective that AT(X(f)) overfits to X(f).
>
> Thus, we suggest the following framing for our paper: what should we use as X(f) (i.e., attack model) such that AT(X(f)) yields models that are robust to a broad classes of physically realizable attacks (rather than just X(f))?
>
> We demonstrate that
>
> a) X(f) should not be l-p norm constrained attacks (for p = infinity or 2, through evaluation with X(f) = PGD and randomized smoothing defense), and
>
> b) Our proposed abstract ROA model appears to be an effective X(f) if we care about physical attacks involving masking/occlusions (Section 5, which evaluates on attacks that are very different from ROA).
>
> Given this preamble, we are now ready to address the specific comments made by the reviewer.
>
> 1) Comment: "the two physical attacks evaluated in this paper have similar attacking patterns, i.e., mask-based pixel attacks. So it is not surprising that DOA is more robust in these cases since DOA is trained on this attacking pattern."
>
> Response:
>
> In fact, our threat model is physical attacks that are mask-based---we call these "adversarial occlusion attacks" (see Section 4).  We grant that this does not capture all conceivable physical attacks.  Our argument is that it captures a very important class of them: physical attacks in which a physical object, such as a stop sign, is physically modified in a manner that is unsuspicious to human observers.
>
> We would dispute, however, the assertion that "DOA is trained on this attacking pattern".  DOA is trained on a very specific pattern---a single rectangle.  Our evaluation in Section 5, however, uses a host of very different patterns, such as eyeglass frames in the face recognition attack, and collections of disjoint rectangles in the stop sign attack (the disjoint part is important, since a union of rectangular occlusions is quite different from a single rectangular occlusion).
>
> 2) Comment: "Actually it has been shown that the framework of adversarial training (AT) will overfit to the attacking patterns used in training. For example, PGD-AT models are less robust to simple non-pixel transformation, like rotation, than the standard models. So what DOA does is just substituting the PGD module in AT to overfit the new attacking patterns, which is of limited contribution and novelty."
>
> Response:
>
> This is indeed conventional thinking in AML, but our higher-level purpose behind this work is very much to contest it.  In particular, we argue that the illustrations such as failure of AT(PGD) on, say, rotation attacks and (as we demonstrate) physical attacks is not a failure of AT(X(f)) in general, but a failure *specifically* of AT(PGD) (i.e., PGD being the wrong attack model to use).
>
> [1] provides additional evidence for this argument in the context of malware detection.  Specifically, the relevant observation in [1] is that for some attack models X(f), AT(X(f)) fails to generalize to other attacks, whereas for the "correct" X(f) (in that case, it actually happens to be a variation of l-p attacks, but not exactly PGD), AT(X(f)) indeed exhibits generalized robustness.
>
> This is why substituting PGD with something else (i.e., ROA) is actually the crucial point, and not merely rehashing common issues.  The central purpose of Section 5 is then to show that AT(ROA) indeed generalizes to physical masking (occlusion) attacks that use entirely different patterns (eyeglass frames, unions of rectangles which yield complex graffiti-looking shapes).
>
> [1] Tong, et al. Improving robustness of ML classifiers against realizable evasion attacks using conserved features.  USENIX Security, 2019.

---

### Author Response · Authors · 2019-11-08
**Response to all: we have uploaded a revised draft**

We are grateful to all the reviewers for their thoughtful comments.

We have made some changes to the draft in response.  The revised draft has been uploaded (please see the latest uploaded draft).

---

### Decision · Program_Chairs · 2019-12-19

**Decision:**

Accept (Spotlight)

**Comment:**

This paper studies the problem of defending deep neural network approaches for image classification from physically realizable attacks. It first demonstrates that adversarial training with PGD attacks and randomized smoothing exhibit limited effectiveness against three of the highest profile physical attacks. Then, it proposes a new abstract adversarial model, where an adversary places a small adversarially crafted rectangle in an image, and develops two approaches for efficiently computing the resulting adversarial examples. Empirical results show the effectiveness. Overall, a good paper. The rebuttal is convincing.